# EEGPT: Pretrained Transformer for Universal and Reliable Representation of EEG Signals

**Guagnyu Wang**
Faculty of Computing
Harbin Institute of Technology
wangguangyu@stu.hit.edu.cn

**Wenchao Liu**
Faculty of Computing
Harbin Institute of Technology
23b903096@stu.hit.edu.cn

**Yuhong He**
Faculty of Computing
Harbin Institute of Technology
19S003002@stu.hit.edu.cn

**Cong Xu**
Faculty of Computing
Harbin Institute of Technology
congxu@hit.edu.cn

**Lin Ma**
Faculty of Computing
Harbin Institute of Technology
malin_li@hit.edu.cn

**Haifeng Li**[*]
Faculty of Computing
Harbin Institute of Technology
lihaifeng@hit.edu.cn

## Abstract

Electroencephalography (EEG) is crucial for recording brain activity, with applications in medicine, neuroscience, and brain-computer interfaces (BCI). However, challenges such as low signal-to-noise ratio (SNR), high inter-subject variability, and channel mismatch complicate the extraction of robust, universal EEG representations. We propose EEGPT, a novel 10-million-parameter pretrained transformer model designed for universal EEG feature extraction. In EEGPT, a mask-based dual self-supervised learning method for efficient feature extraction is designed. Compared to other mask-based self-supervised learning methods, EEGPT introduces spatio-temporal representation alignment. This involves constructing a self-supervised task based on EEG representations that possess high SNR and rich semantic information, rather than on raw signals. Consequently, this approach mitigates the issue of poor feature quality typically extracted from low SNR signals. Additionally, EEGPT's hierarchical structure processes spatial and temporal information separately, reducing computational complexity while increasing flexibility and adaptability for BCI applications. By training on a large mixed multi-task EEG dataset, we fully exploit EEGPT's capabilities. The experiment validates the efficacy and scalability of EEGPT, achieving state-of-the-art performance on a range of downstream tasks with linear-probing. Our research advances EEG representation learning, offering innovative solutions for bio-signal processing and AI applications. The code for this paper is available at: https://github.com/BINE022/EEGPT.

## 1 Introduction

Electroencephalography (EEG) dynamically reflects the brain's functional state by recording electrical signals from the cerebral cortex [1]. EEG is essential for studying brain activity and is pivotal in

---

[*]Corresponding author

brain-computer interface (BCI) applications due to its non-invasive and portable nature [2]. Despite its potential, EEG-based methods face practical challenges due to low signal-to-noise ratio (SNR) [3], high inter-subject variability, and significant task-dependent variations in EEG signals [4]. Self-supervised learning, as described by Yann LeCun in his AAAI 2020 keynote [5], has shown advantages in natural language processing (NLP) [6], computer vision (CV) [7, 8], and speech analysis [9]. More and more state-of-the-art (SOTA) models are pretrained by self-supervised learning on large datasets and fine-tuned for specific applications, effectively reducing the need for extensive labeled data. Masked autoencoders, a type of self-supervised learning method, have been successful in NLP [6] and CV [8] by recovering masked patches based on context.

Recent advances in EEG analysis using self-supervised learning techniques have shown promising results. In Falck et al. [10], a framework for learning EEG representations through contrastive learning was proposed. This framework extends the SimCLR framework to time series data, training a channel feature extractor. The model achieved an accuracy of 85.12% on the Sleep-EDF dataset [11] but was only trained on EEG data collected during 20-second tasks, making it unsuitable for shorter tasks such as motor imagery. BENDR [12] applied self-supervised learning based on masked autoencoders and contrastive learning to EEG data. This approach addresses the challenges of multi-task and multi-paradigm EEG data characterization, enhancing the model's universality. BENDR uses a convolutional encoder to extract features from local time windows, masks some features, and then a transformer decoder predicts the information in the masked parts. EEG2VEC [13] introduced a self-supervised model that learns EEG representations based on contrastive loss and reconstruction loss. The pretrained model is used as a feature extractor for downstream tasks. Both EEG2VEC and BENDR utilize convolutional neural network and transformer networks to learn local and global features. EEG2VEC was validated in EEG match-mismatch and EEG regression tasks of the auditory EEG challenge [14]. The Biosignal Transformer (BIOT) [15] model addresses the challenges of cross-data learning with mismatched channels, variable lengths, and missing values in biosignals such as EEG, ECG, and human activity sensory signals. BIOT tokenizes each channel separately into fixed-length segments containing local signal features and then re-arranges the segments to form a long "sentence". In the CHB-MIT seizure detection task [16], the pretrained BIOT models achieved a 4% improvement. The Large Brain Model (LaBraM) [17] addresses EEG-based deep learning model limitations by enabling cross-dataset learning. It segments EEG signals into channel patches and uses vector-quantized neural spectrum prediction for training a neural tokenizer. This tokenizer encodes raw EEG patches into neural codes, which pretrain transformers to predict the original neural codes for masked patches. LaBraM outperformed SOTA methods in abnormal detection, event type classification [18], emotion recognition [19], and gait prediction [20].

Universal models have clearly made progress in EEG data analysis. However, the extremely low SNR of EEG signals and the complexity of brain activities during tasks make it challenging to learn abstract features using masked autoencoders, which are commonly used in NLP and CV [12]. Additionally, the inconsistent sampling rates of different EEG acquisition devices and variations in electrode channel locations [15] hinder the convolutional encoder's ability to decouple the correlation between electrode channels and EEG signals, resulting in robustness and scalability issues.

To address these issues, we propose a dual self-supervised EEG universal representation method based on the spatio-temporal consistency of EEG signals [21], introducing the EEG Pretrained Transformer (EEGPT) for efficient feature extraction. Our method includes spatio-temporal representation alignment and mask-based reconstruction, enhancing representation quality and model robustness. Our method adopts a BERT-style masked recovery task [22] (not autoregressive task) objectives for pretraining. Beyond the original waveform recovery, we align the predicted EEG signal features of the masked parts with full EEG signal features. Additionally, a local spatio-temporal embedding method improves compatibility across different EEG acquisition devices.

EEGPT, with over 10 million parameters, is pretrained on a mixed multi-task EEG dataset, including data from PhysioMI [23], HGD [24], and M3CV [25]. This pretraining enables EEGPT to extract universal representations for various tasks. For downstream tasks, we employ a linear-probing method, which achieves SOTA performance while also reducing computational resource consumption and preventing overfitting. Our experiments demonstrate EEGPT's superior performance in motor imagery classification [26], event-related potential (ERP) detection [27], and sleep stage detection [28], showcasing its capability to extract high-level abstract features across spatio-temporal dimensions.

Contributions of this paper:

- Proposal of EEGPT, a 10-million-parameter model for EEG universal feature extraction, leveraging a mixed dataset to enhance performances across tasks and subjects.
- Development of a dual self-supervised method for EEG signals, combining spatio-temporal representation alignment and mask-based reconstruction, improving feature quality and convergence.
- Design of a hierarchical structure for decoupled processing of spatial and temporal information, reducing computational complexity and enhancing model flexibility for BCI applications.
- Implementation of a local spatio-temporal embedding method, increasing robustness and compatibility across different EEG acquisition devices.
- Conduct of comprehensive experiments on downstream datasets, demonstrating EEGPT significantly outperforms existing models across multiple EEG tasks and that larger models exhibit improved performance.

## 2 Method

**Background:** According to Kong and Zhang [29], a masked autoencoder learns features through a form of denoising autoencoder: input signals occluded with random patch masks are fed into the encoder, and the decoder predicts the original embeddings of the masked patches:

$$\min_{\theta,\phi} \mathop{\mathbb{E}}_{x \sim \mathcal{D}} \mathcal{H}\left(d_\phi(z), x \odot (1 - \mathsf{M})\right), \quad z = f_\theta(x \odot \mathsf{M}) \tag{1}$$

where "$\odot$" denotes element-wise product; $\mathsf{M}$ is the patch mask; $f_\theta(\cdot)$ and $d_\phi(\cdot)$ are the encoder and decoder, respectively; $z$ is the learned representation; and $\mathcal{H}(\cdot, \cdot)$ is the similarity measurement. By minimizing the loss function, the model learns the optimal representation $z$ of the input signal. However, in practice, there is no explicit representation $z$ (no split of encoder and decoder) in the BERT-style model [22], and the model must be fine-tuned to locate effective representations. In contrast, we add a spatio-temporal representation alignment branch to explicitly represent $z$, which changes Equation 1 to Equation 2 (dual self-supervised method):

$$\min_{\theta,\phi} \mathop{\mathbb{E}}_{x \sim \mathcal{D}} \mathcal{H}\left(d_\phi(z), x \odot (1 - \mathsf{M})\right) + \mathcal{H}(z, f_\theta(x)), \quad z = f_\theta(x \odot \mathsf{M}) \tag{2}$$

This method encourages the encoded representations to take on a larger extent of semantics, similar to the minimal sufficient representation in the Multi-View Entropy Bottleneck (MVEB) approach [30], thereby improving the encoding quality and generalization [31].

**EEG Pretrained Transformer (EEGPT):** The structure of the EEGPT model is shown in Figure 1, which includes operations such as patching, embedding, masking, encoder, predictor and reconstructor. First, the model chunks the input EEG signal $x \in \mathbb{R}^{M \times T}$ ($M$ channels and $T$ time points) into patches $p_{i,j}$, and embeds each patch as a token $token_{i,j}$ by local spatio-temporal embedding (Section 2.3), followed by splitting into masked parts $\mathcal{M}$ and unmasked parts $\overline{\mathcal{M}}$, respectively. Then, we pretain the model using a dual self-supervised learning method, including spatio-temporal representation alignment (Section 2.1) and mask-based reconstruction (Section 2.2). Finally, the linear-probing method is used in downstream tasks (Section 2.4).

### 2.1 Spatio-temporal Representation Alignment

The spatio-temporal representation alignment method aligns the predicted features with the momentum encoder's output, enhancing the encoder's ability to extract robust features and ensuring that the encoder's output contains high-quality global features. We employ the encoder and the predictor to decouple spatial and temporal features, significantly reducing computational complexity.

**Encoder:** The encoder integrates spatial information from masked patches. Equation 3 describes how the encoder (ENC) processes all masked $token_{i,j}$ at time $j$ as input and produces the corresponding output feature $enc_j$:

$$enc_j = \mathrm{ENC}\left( \{token_{i,j}\}_{(i,j)\in\mathcal{M}} \right) \tag{3}$$

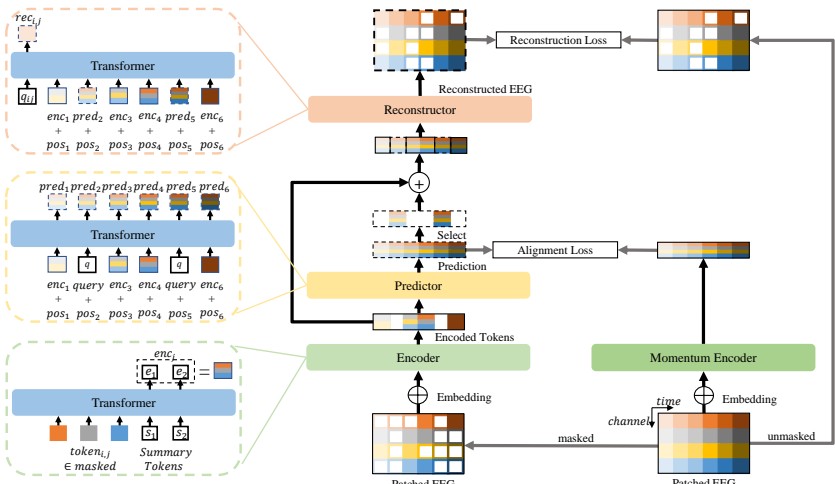

Figure 1: The EEGPT structure involves patching the input EEG signal as $p_{i,j}$ through masking (50% time and 80% channel patches), creating masked part $\mathcal{M}$ and unmasked part $\overline{\mathcal{M}}$ and then embedding as $token_{i,j}$ by local spatio-temporal embedding. The encoder processes the masked part, extracting features ($enc_j$) consisting of $\{e_i\}_{i=1}^{S}$ for each time segment in the $\mathcal{M}$ part with summary tokens $\{s_i\}_{i=1}^{S}$. The predictor predicts features ($pred_j$) for all time segments, aligning with the Momentum Encoder output ($menc_j$). Based on features extracted by the predictor and encoder, the reconstructor generates $rec_{i,j}$ to reconstruct the EEG signal of the $\overline{\mathcal{M}}$ part.

**Predictor:** As in Equation 4, the predictor (PRED) utilizes features $enc_j$ of the masked part from the encoder, combined with temporal position information $pos_j$, to predict the complete encoded feature. Embedding and unembedding [32] are performed linearly on the input and output. We adopt the rotary position embedding method [33] to generate $pos_j$, introducing relative positional and temporal information. To generate prediction features belonging to $\overline{\mathcal{M}}$, a learnable vector $query$ is used as the query token. Through self-supervised training, the encoder is encouraged to extract more information about the correlation among tokens:

$$\{pred_t\}_{t\in\{1,2,...,N\}} = \text{PRED}\left(\{enc_j + pos_j\}_{\exists i,(i,j)\in\mathcal{M}}\right) \qquad (4)$$

**Momentum Encoder:** The structure of the momentum encoder is identical to that of the encoder. Equation 5 outlines how the momentum encoder (MENC) processes all $token_{i,j}$ at time $j$ as input and produces the corresponding output $menc_j$. After each training iteration, the parameters of the encoder are accumulated into the momentum encoder with a factor of $\tau = 0.01$.

$$menc_j = \text{MENC}\left(\{token_{i,j}\}_{(i,j)\in\mathcal{M}\cup\overline{\mathcal{M}}}\right) \qquad (5)$$

We employ an alignment loss based on Mean Square Error (MSE) [34] to achieve spatio-temporal representation alignment:

$$\mathcal{L}_A = -\frac{1}{N}\sum_{j=1}^{N}||pred_j, \text{LN}(menc_j)||_2^2 \qquad (6)$$

In Equation 6, LN denotes layer normalization [35], which helps to mitigate the effects of extreme values and covariate shift, allowing the model to focus on more important features.

## 2.2 Mask-based Reconstruction

The mask-based reconstruction method aligns the reconstructed patches generated by the reconstructor with the raw patches $p_{i,j}$ in the $\overline{\mathcal{M}}$ part.

**Reconstructor:** As shown in Equation 7, the reconstructor (REC) utilizes features $enc_j$ from the $\mathcal{M}$ part encoded by the encoder and features $pred_j$ of the $\overline{\mathcal{M}}$ part predicted by the predictor, along with temporal position $pos_j$, to generate the reconstructed patch $rec_{u,t}$. We establish a "skip connection" between the encoder and the reconstructor to help maintain the features and accelerate convergence.

$$\underset{(u,t)\in\overline{\mathcal{M}}}{\{rec_{u,t}\}} = \text{REC}\left(\underset{\exists i,(i,j)\in\mathcal{M}}{\{enc_j + pos_j\}} \cup \underset{\forall i,(i,j)\in\overline{\mathcal{M}}}{\{pred_j + pos_j\}}\right) \tag{7}$$

Mask-based reconstruction is achieved using a reconstruction loss based on the Mean Square Error (MSE):

$$\mathcal{L}_R = -\frac{1}{|\mathcal{M}|} \sum_{(i,j)\in\overline{\mathcal{M}}} ||rec_{i,j}, \text{LN}(p_{i,j})||_2^2 \tag{8}$$

The complete pretraining loss $\mathcal{L}$ is constructed by summing both $\mathcal{L}_A$ and $\mathcal{L}_R$:

$$\mathcal{L} = \mathcal{L}_A + \mathcal{L}_R \tag{9}$$

## 2.3 Local Spatio-Temporal Embedding

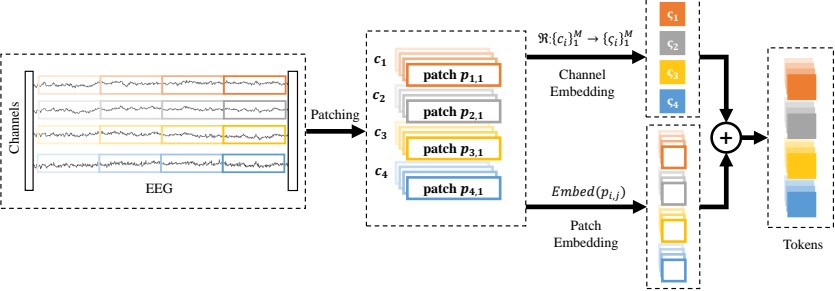

Figure 2: Illustration of local spatio-temporal embedding. The EEG signal is divided into equally sized patches in the spatio-temporal dimensions. Each patch represents a time segment for a specific channel without overlap. The patches are linearly embedded and incorporated with channel embedding information to obtain a corresponding feature.

The local spatio-temporal embedding method first patches and embeds the EEG signal in the spatio-temporal dimension before feeding it into the encoder, as shown in Figure 2. We denote the set of EEG signal channels as $\{c_i\}_{i=1}^M$, where $c_i$ is the name of each channel. Firstly, the EEG signal is divided into equally sized patches in the spatio-temporal dimensions, denoted as $p_{i,j}, i \in \{1, 2, ..., M\}, j \in \{1, 2, ..., N\}$:

$$p_{i,j} = x_{i,(j-1)d:jd} \tag{10}$$

where $d$ represents patch's time length, and $N = T/d$ is the number of time patches. Next, the patches are linearly embedded, combining the channel embedding information. We construct a Codex book [36] $\{\varsigma_i \in \mathbb{R}^{d_e}\}_{i=1}^M$ ($d_e$ is the embedding dimension) containing all learnable channel embedding vectors and a mapping from channel names to channel embedding vectors $\Re : \{c_i\}_{i=1}^M \to \{\varsigma_i\}_{i=1}^M$. This mapping flexibly corresponds the channels of the EEG data to the channels of the model inputs, allowing the model to adapt to multiple datasets and improve channel adaptation. The patch's content is linearly embedded as $Embed(p_{i,j}) = W_p^T p_{i,j} + b_p$, where $W_p \in \mathbb{R}^{d \times d_e}$ and $b_p \in \mathbb{R}^{d_e}$ are learnable parameters. Embedded token denotes as $token_{i,j} \in \mathbb{R}^{d_e}$:

$$token_{i,j} = Embed(p_{i,j}) + \varsigma_i \tag{11}$$

Based on self-supervised learning task, the extracted features of patches are mutually predictable and can ignore noise signals at smaller scales. The method aims to extract macroscopic features that span larger scales, which are believed to be more easily recognizable and considered meaningful features.

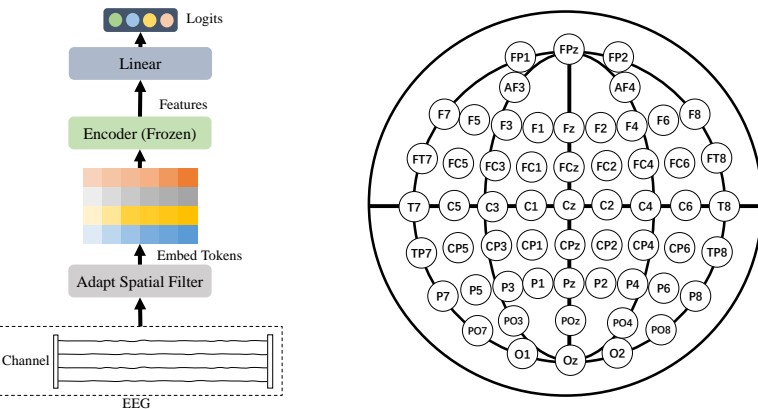

Figure 3: Linear-probing method.       Figure 4: Electrode locations.

## 2.4   Linear-Probing Method

In the downstream tasks, we apply the pretrained encoder and concatenate additional modules to solve classification tasks. As shown in Figure 3, we introduce the linear-probing method, which freezes the parameters in the pretrained model and only changes the parameters in the additional linear modules. These modules include adaptive spatial filters ($1 \times 1$ convolution) for aligning channels between EEG and the model, and a linear layer to map the features to logits. The encoder passes the output tokens corresponding to summary tokens to the linear classification head. This approach helps avoid the overfitting problem when a large parameter model is fine-tuned using a limited number of samples. Since the additional module is straightforward, the performance is solely determined by the encoder, allowing us to assess the model's capability.

# 3   Experiments

## 3.1   Datasets and Data Processing

Table 1: Datasets for pretraining and downstream tasks

|                      | Datasets   | Paradigms | Subjects | Targets |
|----------------------|------------|-----------|----------|---------|
| pretraining Datasets | PhysioMI   | MI&ME     | 109      | 5       |
|                      | HGD        | MI        | 14       | 4       |
|                      | TSU        | SSVEP     | 35       | 40      |
|                      | SEED       | EMO       | 15       | 3       |
|                      | M3CV       | MULTI     | 106      | -       |
| Downstream Datasets  | BCIC-2A    | MI        | 10       | 4       |
|                      | BCIC-2B    | MI        | 10       | 2       |
|                      | Sleep-EDFx | SLEEP     | 197      | 5       |
|                      | KaggleERN  | ERN       | 26       | 2       |
|                      | PhysioP300 | P300      | 9        | 2       |
|                      | TUAB       | Abnormal  | 2383     | 2       |
|                      | TUEV       | Event     | 288      | 6       |

We curated EEG public datasets of various paradigms for model pretraining as shown in Table 1. It contains motor imagery (MI) and execution (ME) datasets PhysioMI [23], HGD [24], steady-state visual evoked potential (SSVEP) dataset TSU [37], emotional classification dataset SEED [38] and multi-subject multi-session multi-paradigm dataset M3CV [25]. To assess the practical utility of the learned representations in downstream tasks, we curated a list of datasets as shown in Table 1. It contains MI datasets BCIC-2A [39], BCIC-2B [40], sleep stage detection dataset Sleep-EDFx [11] , error related negativity (ERN) dataset KaggleERN [41] and event-related potentials dataset PhysioP300 [23]. We also curated a dataset of abnormal EEG signals dataset TUAB and event type classification dataset TUEV from Temple University EEG Corpus [18]. These diverse datasets

enable comprehensive evaluation of the proposed EEGPT model across various tasks. Each dataset underwent similar and distinct preprocessing steps, including cropping (4s), re-referencing (average), channels selecting, scaling (mV) and resampling (256Hz). There are 0-38Hz bandpass filtering in MI datasets for downstream tasks. More details refer to Appendix C.

## 3.2 Implementation and Settings

**Model implementation.** The implementation of encoder, predictor and reconstructor in EEGPT adopts the vision transformer (VIT) [42] and sets $S$ learnable summary tokens (similar to [CLS] token) for summarizing information within the same time patches. We used 58 electrodes ($M = 58$), as shown in Figure 4. The input signal has a sampling rate of $f_s = 256$ Hz, and the input signal time length is $T = 1024$. Each patch has a time length of $d = 64$, corresponding to a 250ms time window. The 50% time and 80% channel patches of the patches are masked during training.

**pretraining strategy.** In pretraining, for each training dataset, we randomly sampled 10% of the samples as the validation set. As shown in Table 6, we trained 8 variants with different embedding dims, layers of transformer models (encoder, predictor, reconstructor) and summary tokens $S$. The AdamW optimizer was employed with the OneCycle learning rate strategy [43] (initial learning rate of 2.5e-4, maximum of 5e-4, minimum of 3.13e-5). The training was conducted for 200 epochs, with a batch size of 64 and 16-bit mixed precision training on 8 Nvidia 3090 GPUs.

**Evaluation strategy.** For the data splitting of TUAB and TUEV, we strictly follow the same strategy as BIOT [15] to compare all methods fairly. In other downstream tasks, we use the same experimental configuration as BENDR [12], in which Leave-One-Subject-Out (LOSO) validation method are used. Specially, KaggleERN uses 4-fold cross-validation with 10 subjects for testing, Sleep-EDFx uses 10-fold cross-validation and ratio 6:2:2 in training, validation and test splitting. We use linear-probing method for downstream tasks. Particularly, in the sleep stage detection task, we use a 4-layer transformer encoder model as a classifier that integrates the output of our model for every 0.25s for the purpose of processing a long task of 30s. We used the optimal large model in Section 3.5 for testing on all downstream tasks. To ensure the reliability of the experiments, we repeated each experiment three times and calculated the standard deviation.

**Baselines & Metrics.** For the TUAB and TUEV datasets, we use the same baselines from BIOT which are fully fine-tuned models. In other tasks, we use the pretrained BENDR [12], BIOT [15] and LaBraM [17] as the baselines. The following metrics are used for comparison: 1) Balanced Accuracy (BAC), 2) AUROC, 3) Weighted F1, 4) Cohen's Kappa. We use AUROC only for binary classification tasks and Weighted F1 only for multi-class classification tasks. More details refer to Appendix D.

## 3.3 Downstream Experiment Results

Table 2: The results of different methods on TUAB.

| Methods | Model Size | Balanced Accuracy | AUROC |
|---|---|---|---|
| SPaRCNet [44] | 0.79M | 0.7896±0.0018 | 0.8676±0.0012 |
| ContraWR [45] | 1.6M | 0.7746±0.0041 | 0.8456±0.0074 |
| CNN-T [46] | 3.2M | 0.7777±0.0022 | 0.8461±0.0013 |
| FFCL [47] | 2.4M | 0.7848±0.0038 | 0.8569±0.0051 |
| ST-T [48] | 3.5M | 0.7966±0.0023 | 0.8707±0.0019 |
| BIOT [15] | 3.2M | 0.7959±0.0057 | 0.8815±0.0043 |
| Ours-Tiny | 4.7M | 0.7959±0.0021 | 0.8716±0.0041 |
| Ours | 25M | 0.7983±0.0030 | 0.8718±0.0050 |

We conducted a comparative analysis with other large models on the Temple University datasets, using the same configuration as BIOT [15] for experiments on the TUAB and TUEV datasets. The results are presented in Tables 2 and 3. In the TUAB dataset, the performance of EEGPT is comparable to that of the BIOT model. In the TUEV dataset, EEGPT improves the balanced accuracy by 9.5%, and the weighted F1 score by 6.9% compared to BIOT.

To further validate the effectiveness of our model, we conducted comparative experiments with BENDR, BIOT, and LaBraM. The results are shown in Table 4. On the BCIC-2A and BCIC-2B datasets for motor imagery tasks and the Sleep-EDFx dataset for sleep stage detection, our

Table 3: The results of different methods on TUEV.

| Methods | Model Size | Balanced Accuracy | Weighted F1 | Cohen's Kappa |
|---|---|---|---|---|
| SPaRCNet [44] | 0.79M | 0.4161±0.0262 | 0.7024±0.0104 | 0.4233±0.0181 |
| ContraWR [45] | 1.6M | 0.4384±0.0349 | 0.6893±0.0136 | 0.3912±0.0237 |
| CNN-T [46] | 3.2M | 0.4087±0.0161 | 0.6854±0.0293 | 0.3815±0.0134 |
| FFCL [47] | 2.4M | 0.3979±0.0104 | 0.6783±0.0120 | 0.3732±0.0188 |
| ST-T [48] | 3.5M | 0.3984±0.0228 | 0.6823±0.0190 | 0.3765±0.0306 |
| BIOT [15] | 3.2M | 0.5281±0.0225 | 0.7492±0.0082 | 0.5273±0.0249 |
| Ours-Tiny | 4.7M | 0.5670±0.0066 | 0.7535±0.0097 | 0.5085±0.0173 |
| Ours | 25M | 0.6232±0.0114 | 0.8187±0.0063 | 0.6351±0.0134 |

Table 4: The results of universal EEG models on various datasets.

| Datasets | Methods | Balanced Accuracy | Cohen's Kappa | Weighted F1 / AUROC |
|---|---|---|---|---|
| BCIC-2A | BENDR | 0.4899±0.0070 | 0.3199±0.0094 | 0.4836±0.0076 |
| | BIOT | 0.4590±0.0196 | 0.2787±0.0261 | 0.4282±0.0289 |
| | LaBraM | 0.5613±0.0052 | 0.4151±0.0069 | 0.5520±0.0052 |
| | Ours | **0.5846±0.0070** | **0.4462±0.0094** | **0.5715±0.0051** |
| BCIC-2B | BENDR | 0.7067±0.0011 | 0.4131±0.0022 | 0.7854±0.0029 |
| | BIOT | 0.6409±0.0118 | 0.2817±0.0236 | 0.7095±0.0141 |
| | LaBraM | 0.6851±0.0063 | 0.3703±0.0125 | 0.7576±0.0067 |
| | Ours | **0.7212±0.0019** | **0.4426±0.0037** | **0.8059±0.0032** |
| Sleep-EDFx | BENDR | 0.6655±0.0043 | 0.6659±0.0043 | 0.7507±0.0029 |
| | BIOT | 0.6622±0.0013 | 0.6461±0.0017 | 0.7415±0.0010 |
| | LaBraM | 0.6771±0.0022 | 0.6710±0.0006 | 0.7592±0.0005 |
| | Ours | **0.6917±0.0069** | **0.6857±0.0019** | **0.7654±0.0023** |
| KaggleERN | BENDR | 0.5672±0.0020 | 0.1461±0.0037 | 0.6030±0.0044 |
| | BIOT | 0.5118±0.0089 | 0.0297±0.0224 | 0.5495±0.0167 |
| | LaBraM | 0.5439±0.0029 | 0.0944±0.0066 | 0.5693±0.0052 |
| | Ours | **0.5837±0.0064** | **0.1882±0.0110** | **0.6621±0.0096** |
| PhysioP300 | BENDR | 0.6114±0.0118 | 0.2227±0.0237 | 0.6588±0.0163 |
| | BIOT | 0.5485±0.0325 | 0.0968±0.0647 | 0.5308±0.0333 |
| | LaBraM | 0.6477±0.0110 | 0.2935±0.0227 | 0.7068±0.0134 |
| | Ours | **0.6502±0.0063** | **0.2999±0.0139** | **0.7168±0.0051** |

model exhibited accuracy improvements of 9.4%, 1.5%, 2.6%, respectively, compared to BENDR. Considering that BENDR used full model fine-tuning while our model only fine-tuned an additional linear layer, this suggests that our model extracts richer and more universal features. We used the linear-probing method for BIOT and LaBraM. EEGPT improved by 2.3%, 3.6%, and 1.4%, respectively, compared to LaBraM, and by 12.5%, and 8.1%, and 2.9%, respectively, compared to BIOT. For the ERP-type task datasets, our model outperforms BENDR by 2.6% and 3.9% on the KaggleERN and PhysioP300 datasets, respectively. The performance of our model on the PhysioP300 dataset is comparable to that of LaBraM, but higher by 3.0% on KaggleERN. Our model also outperforms BIOT on the KaggleERN and PhysioP300 by 7.2% and 10.2%, respectively. On all tasks, our model EEGPT achieves competitive results compared to BENDR, BIOT, and LaBraM. This demonstrates that EEGPT learns consistent representational features over the temporal-spatial dimensions, enabling the model to be more widely applied to multiple paradigm tasks and to achieve better classification performance.

Combining the above experimental results, we demonstrate that the method proposed in this paper addresses the issues of poor EEG channel adaptability, poor quality of EEG representations extracted by existing self-supervised learning methods, and the lack of universality of representations across multiple paradigms. Our method effectively extracts high-quality universal EEG representations.

## 3.4 Ablation Experiment Results

We conducted ablation experiments using the large model with four different configurations, and the results are presented in Table 5. In the absence of alignment loss ($\mathcal{L}_A$), the reconstruction loss

Table 5: The results of the ablation study.

| Variants | $\mathcal{L}_A$ | $\mathcal{L}_R$ | BCIC-2A-BAC | BCIC-2B-AUROC | KaggleERN-AUROC |
|---|---|---|---|---|---|
| A: w/o $\mathcal{L}_A$ | 37.13 | 0.57 | 0.5287±0.0086 | 0.7264±0.0381 | 0.5752±0.0164 |
| B: w/o LN | 0.15 | 0.002 | 0.5567±0.0088 | 0.7920±0.0012 | 0.5891±0.0227 |
| C: w/o skip | 0.12 | 0.56 | 0.5796±0.0011 | 0.7702±0.0122 | 0.6356±0.0296 |
| D: with all | 0.24 | 0.56 | **0.5846±0.0070** | **0.8059±0.0032** | **0.6621±0.0096** |

Table 6: The results of pretrained models.

| variants | $d_e$ | layers | $S$ | params | $\mathcal{L}_A$ | $\mathcal{L}_R$ | BCIC-2A-BAC (%) |
|---|---|---|---|---|---|---|---|
| tiny1 | 64 | 2/2/4 | 1 | 0.4M | 0.32 | 0.60 | 49.19 |
| tiny2 | 64 | 2/2/4 | 4 | 0.5M | 0.36 | 0.60 | 50.03 |
| tiny3 | 64 | 8/8/8 | 4 | 1.6M | 0.17 | 0.59 | 51.58 |
| little | 128 | 8/8/8 | 4 | 6.4M | 0.18 | 0.57 | 54.18 |
| base1 | 256 | 6/6/6 | 1 | 19M | 0.24 | 0.56 | 54.53 |
| base2 | 256 | 8/8/8 | 4 | 25M | 0.33 | 0.56 | 56.48 |
| base3 | 512 | 6/6/6 | 1 | 76M | 0.14 | 0.58 | 54.47 |
| large | 512 | 8/8/8 | 4 | 101M | 0.24 | 0.56 | **58.46** |

is comparable to that of the version D model, but there is a significant performance degradation of 6%∼9% in the downstream task. Without layer normalization on the targets of reconstruction loss, the version B model demonstrated a lower pretrain loss $\mathcal{L}$, but it was affected by extreme values and covariate shift [35], resulting in a 3%,1% and 7% reduction in downstream tasks performance. The version C model, which removes the skip connection and uses all $\{pred_t\}_{t=1}^N$ from the predictor as inputs to the reconstructor, exhibited lower alignment loss and comparable reconstruction loss compared to the version D model, but showed a 1%∼3% lower performance in the downstream task. These results suggest that the dual self-supervised method proposed in this paper is effective, as the spatio-temporal alignment improves the quality of the EEG representations extracted by the model.

## 3.5 Pretrain Experiment Results

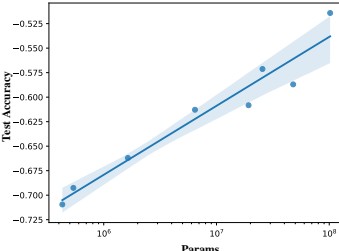 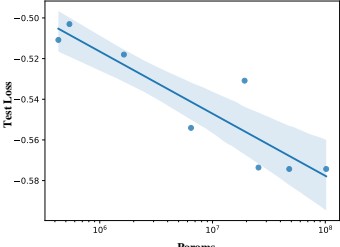

Figure 5: Scaling laws with EEGPT parameter size N. Axes are all on a logarithmic scale.

We designed 8 variants to investigate the effects of model size (embedding size $d_e$ / layers) and summary token ($S$) on pretraining loss and downstream task accuracy. To evaluate the performance of the pretrained models, we used the linear-probing method to test cross-subject classification tasks on the BCIC-2A dataset. The experimental results are presented in Table 6. As the model size (embedding size $d_e$ / layers) and summary tokens ($S$) increased, the reconstruction loss ($\mathcal{L}_R$) gradually decreased, and the performance on the downstream task improved. For models with the same $S$, larger models exhibited lower alignment loss $\mathcal{L}_A$ and higher performance. We also investigated the trends in accuracy and reconstruction loss as the model size increased, as shown in Figure 5. These trends can be summarized by the scaling laws: $\text{ACC} = (33.6 * N)^{0.029}$ and $\mathcal{L}_R = (0.72 * N)^{-0.014}$, where $N$ is the parameter count of the model. The results of the downstream task experiments indicate that larger models generally achieve higher accuracy, with the large model, featuring an 8-layer, 512-embedding dimension, and 4 summary tokens, exhibiting the highest accuracy. For simplicity, we used the optimal large model for testing on all downstream tasks.

# 4 Conclusion

In this paper, we propose a self-supervised EEG Pretrained Transformer (EEGPT) model with over 10 million parameters for universal EEG representation learning. We employ a dual self-supervised approach for pretraining, involving spatio-temporal representation alignment and mask-based reconstruction. The spatio-temporal representation alignment aligns masked patches' features with full patches' features, enhancing the quality of EEG representations and concentrating key information in the encoder output. The mask-based reconstruction leverages the spatial and temporal consistency exhibited by EEG signals to extract complementary features in both dimensions. We design a hierarchical structure for EEGPT, which first extracts stable spatial representations from short-term EEG signals, then captures the temporal correlations among long-term EEG signals. This structure not only reduces computational complexity but also enhances the flexibility and adaptability of EEGPT in BCI applications. Experiments demonstrate that our dual self-supervised pretrain model significantly outperforms the popular universal feature extraction models BENDR, BIOT, and LaBraM on tasks such as motor imagery, sleep stage detection and ERP-type classification. Compared with BIOT on the TUEV dataset, we achieved a 9.5% performance improvement. These tasks, which have different channel configurations and sampling rates, suggest that EEGPT is scalable. In the future, we plan to further enrich the pretraining EEG dataset and expand the model size and applicability.

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

# A    Additional results

This section provides additional experimental results to support the claims in the main paper.

## A.1    Ablation study for the predictor

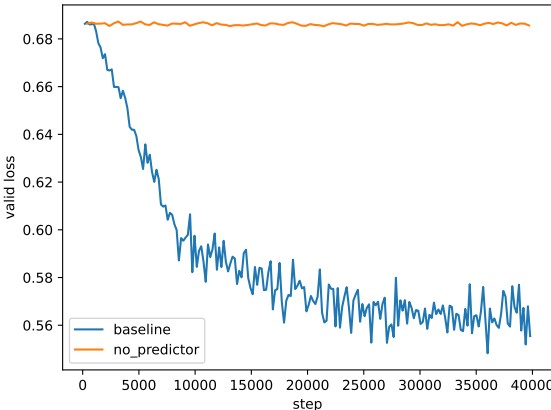

Figure 6: Validation loss curves during pretraining, where 'baseline' is the standard EEGPT model and 'no_predictor' is the model without the predictor.

We conducted a pretraining experiment after removing the predictor, the variation of $L_R$ loss with the number of iteration steps during training is shown in Figure 6. In this figure, 'baseline' is the standard EEGPT model and 'no_predictor' is the model without the predictor. We can see that the reconstruction loss $L_R$ of the 'no_predictor' model after removing the predictor does not decrease, which indicates that directly aligning the outputs of the encoder and the momentum encoder does lead to the problem of representation collapse, resulting in the reconstruction task not being able to learn meaningful representation.

## A.2    Ablation study for pretraining methods

Table 7: The results of the ablation study for pretraining methods.

| Variants | $\mathcal{L}_A$ | $\mathcal{L}_R$ | BCIC-2A-BAC | BCIC-2B-AUROC | KaggleERN-AUROC |
|---|---|---|---|---|---|
| A: w/o $\mathcal{L}_A$ | 37.13 | 0.57 | 0.5287±0.0086 | 0.7264±0.0381 | 0.5752±0.0164 |
| B: w/o LN | 0.15 | 0.002 | 0.5567±0.0088 | 0.7920±0.0012 | 0.5891±0.0227 |
| C: w/o skip | 0.12 | 0.56 | 0.5796±0.0011 | 0.7702±0.0122 | 0.6356±0.0296 |
| D: with all | 0.24 | 0.56 | **0.5846±0.0070** | **0.8059±0.0032** | **0.6621±0.0096** |

In the ablation experiments, we used the BCIC-2A, BCIC-2B and KaggleERN datasets to test the model with the linear probing method, the results are shown in Table 7. We can conclude that (1) the model performance degradation on all datasets without $L_A$ loss is significant (6% to 9%); (2) without layer normalisation on the reconstruction target, the performance of the version B model degraded by 3%, 1%, and 7% on BCIC-2A, BCIC-2B, and KaggleERN, respectively; and (3) the performance of version C model (removed the skip connection) is reduced by 1%, 3% and 3% on these datasets, respectively.

## A.3    Ablation study for fine-tuning methods

We have added experiments comparing linear probing method and full fine-tuning method, as well as experiments comparing with and without adaptive spatial filter. The experimental results are displayed in Table 8. In Table 8, 'ASF' stands for with an adaptive spatial filter, in contrast to feeding the signal directly into the model; 'L-P' stands for using linear probing, in contrast to using full fine-tuning of the model. Model variants A and C are the models with full fine-tuning and linear probing after excluding the adaptive spatial filter, respectively. Model variants B and D are models

Table 8: The results of the ablation study for fine-tuning methods.

| Variants | ASF | L-P | BCIC-2A-BAC | BCIC-2B-AUROC | KaggleERN-AUROC |
|----------|-----|-----|-------------|----------------|------------------|
| A | | | 0.5774±0.0072 | 0.7871±0.0054 | 0.6078±0.0101 |
| B | ✓ | | 0.5183±0.0155 | 0.7541±0.0083 | 0.6110±0.0019 |
| C | | ✓ | 0.5586±0.0089 | 0.7974±0.0030 | 0.6463±0.0081 |
| D | ✓ | ✓ | **0.5846±0.0070** | **0.8059±0.0032** | **0.6621±0.0096** |

with full fine-tuning and linear probing after using adaptive spatial filter, respectively. The results show that on the BCIC-2B and KaggleERN datasets, variants C and D tested with linear probing achieved better results than variants A and B using full fine-tuning; on the BCIC-2A dataset, variant A with full fine-tuning and no adaptive spatial filter is close to variant D with linear probing and an adaptive spatial filter, but overall the linear probing used by models C and D outperform A and B. The results show that variants B and D using adaptive spatial filter achieve better results compared to variants A and C without adaptive spatial filter.

## A.4 Scaling laws experiments

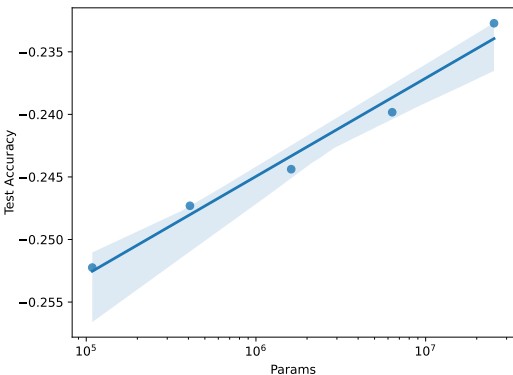

Figure 7: Results on TUAB dataset. Scaling laws with EEGPT parameter size N. Axes are all on a logarithmic scale.

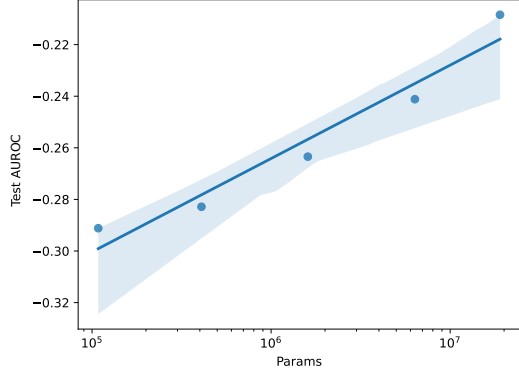

Figure 8: Results on BCIC-2B dataset. Scaling laws with EEGPT parameter size N. Axes are all on a logarithmic scale.

We added the results of the scale law experiments on the TUAB dataset and the BCIC-2B dataset, as shown in Figure 7 and 8. The results on the TUAB dataset show that the scaling law of the test balanced accuracy metric with model size ($N$) for TUAB is: $BAC = (0.74 * N)^{0.0034}$; the results

on the BCIC-2B dataset show that the scaling law of the AUROC metric with model size ($N$) for BCIC-2B is: $\text{AUROC} = (0.61 * N)^{0.0157}$.

## A.5 Effect of pretrain data size

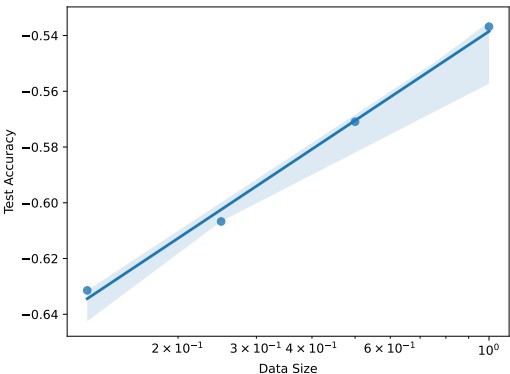

Figure 9: Results on BCIC-2A dataset. Scaling laws with pretrain data size D. Axes are all on a logarithmic scale.

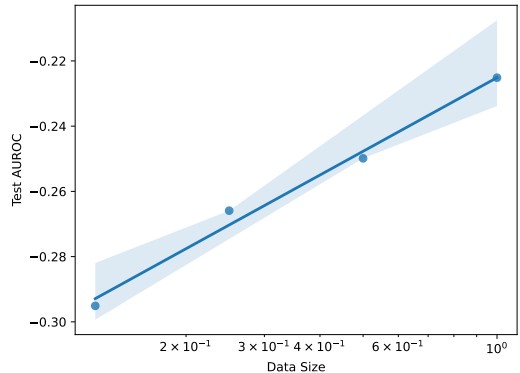

Figure 10: Results on BCIC-2B dataset. Scaling laws with pretrain data size D. Axes are all on a logarithmic scale.

We added pretraining experiments using 100%, 50%, 25%, and 12.5% of the training data and tested them on the downstream tasks of BCIC-2A and BCIC-2B. The results are presented in Figure 9 and 10. The results show that the scaling law of the balanced accuracy metric with the total amount of data $D$ (the percentage of the training data used) for BCIC-2A is: $\text{ACC} = (0.58 * D)^{0.0461}$; and the scaling law of the AUROC metric with the total amount of data $D$ for BCIC-2B is:$\text{AUROC} = (0.79 * D)^{0.0325}$.

## A.6 Results of LaBraM on TUAB and TUEV

Table 9: The results of different methods on TUAB.

| Methods | Model Size | Balanced Accuracy | AUROC |
|---|---|---|---|
| BIOT [15] | 3.2M | 0.7959±0.0057 | 0.8815±0.0043 |
| LaBraM [17] | 5.8M | 0.8140±0.0019 | 0.9022±0.0009 |
| Ours-Tiny | 4.7M | 0.7959±0.0021 | 0.8716±0.0041 |
| Ours | 25M | 0.7983±0.0030 | 0.8718±0.0050 |

Table 10: The results of different methods on TUEV.

| Methods | Model Size | Balanced Accuracy | Weighted F1 | Cohen's Kappa |
|---------|-----------|-------------------|-------------|---------------|
| BIOT [15] | 3.2M | 0.5281±0.0225 | 0.7492±0.0082 | 0.5273±0.0249 |
| LaBraM [17] | 5.8M | 0.6409±0.0065 | 0.8312±0.0052 | 0.6637±0.0093 |
| Ours-Tiny | 4.7M | 0.5670±0.0066 | 0.7535±0.0097 | 0.5085±0.0173 |
| Ours | 25M | 0.6232±0.0114 | 0.8187±0.0063 | 0.6351±0.0134 |

LaBraM employs a larger pretraining dataset than ours, which also contains the TUEG dataset that is similar to the TUAB and TUEV distributions, and whose paper illustrates the scale effect of the amount of pretraining data, which may have facilitated the learning of a richer and more downstream task-adaptable representation for LaBraM.

### A.7    Ablation study of self-supervised learning on TUAB

Table 11: The results of different methods on TUAB.

| Methods | Model Size | Balanced Accuracy | AUROC |
|---------|-----------|-------------------|-------|
| BIOT [15] | 3.2M | 0.7959±0.0057 | 0.8815±0.0043 |
| Ours (no pretrained) | 25M | 0.7553±0.0014 | 0.8260±0.0018 |
| Ours | 25M | 0.7983±0.0030 | 0.8718±0.0050 |

We tested the model with randomly initialized parameters (no pretrained model) on the TUAB dataset and the test results are shown in Table 11. In Table 11, 'Ours (no pretrained)' represents the model that is not loaded with pretrained parameters. By comparison, we see that the model without using self-supervised pretraining has the worst performance on TUAB, with a 4% reduction in balance accuracy and a 5% reduction in AUROC compared to the model loaded with pretraining parameters.

## B    CODE ACCESS

The code for this paper is available at: `https://github.com/BINE022/EEGPT`.

## C    DATASET DESCRIPTION

### C.1    PRETRAINING DATASET DESCRIPTION

#### C.1.1    Motor Imagery and Motor Execution tasks: PhysioNetMI [23]

The PhysioNetMI dataset[2] consists of over 1500 one-minute and two-minute EEG recordings obtained from 109 volunteers. Subjects performed different motor execution/imagery tasks while recording 64-channel EEG using the BCI2000 system.

**Preprocessing:** All eight tasks were used during the pretraining of this paper, using a global average reference, intercepting data from 0 to 6s after the start of each trial, and randomly intercepting data from a 4s time window of these during pretraining.

#### C.1.2    Motor Imagery task: HGD [24]

The HGD dataset, a 128-electrode dataset, was taken from 14 healthy subjects with approximately 1,000 trials each. The four-second executive movement trials were divided into 13 runs. The four types of movements were left-handed movements, right-handed movements, bipedal movements, and rest. The training set consisted of approximately 880 trials for all but the last two runs, and the test set consisted of approximately 160 trials for the last two runs. We access this dataset by MOABB[3].

**Preprocessing:** All four tasks were used in the pretraining process of this paper, first downsampled to 256 Hz and then standardized to mV, using a global average reference to intercept the data from 0

---

[2]https://physionet.org/content/eegmmidb/1.0.0/
[3]https://neurotechx.github.io/moabb/dataset_summary.html

to 10s after the start of each trial, and randomly intercepting the data from a 4-s time window of these during pretraining.

### C.1.3   SSVEP task: TSU [37]

The TSU dataset[4] is a benchmark dataset for brain-computer interfaces based on steady-state visual evoked potentials (TSUBenckmark). This dataset collects SSVEP-BCI recordings from 35 healthy subjects using a brain-computer interface (BCI) speller with 40 characters for experimental data acquisition. SSVEP-BCI recordings were made for 40 characters that flashed at different frequencies (8-15.8 Hz with 0.2 Hz intervals).

**Stimuli:** Each trial began with a visual cue (red square) indicating the target stimulus. The cue appeared on the screen for 0.5 seconds. Subjects were asked to shift their eyes to the target as soon as possible within the cue duration. After the cue was offset, all stimuli began flashing on the screen simultaneously for 5 seconds. After the stimulus offset, the screen was blanked for 0.5 seconds, and then the next trial began, which allowed subjects to have a short rest period between successive trials.

**Signal:** EEG of 35 subjects (64 channels, 250 Hz). Each subject's experiment consisted of 6 blocks. Each block contained 40 trials corresponding to all 40 characters displayed in randomized order. Total: 35 persons x 6 blocks x 40 trials.

**Preprocessing:** All 40 tasks were used in the pretraining process of this paper, first downsampled to 256 Hz, and then standardized to mV, using a globally averaged reference to intercept the data from 0 to 4s after the start of each trial for pretraining.

### C.1.4   Emotion Recognition task: SEED [38]

The SEED dataset, Shanghai Jiao Tong University Emotion EEG dataset (SEED)[5], is an EEG dataset provided by the BCMI lab led by Prof. Baoliang Lu.

**Stimuli:** 15 four-minute long movie clips from six Chinese movies.

**Signal:** EEG (62 channels, 200 Hz) from 15 subjects and eye movement data from 12 subjects. Three experiments were conducted per subject, each approximately one week apart, for a total of 15 subjects x 3 sessions = 45 subjects.

**Scores:** Positive (1), negative (-1), and neutral (0).

**Preprocessing:** All three tasks were used in the pretraining process of this paper, first downsampled to 256 Hz and then standardized to mV, using a global average reference to intercept the data from 0 to 10s after the start of each trial for pretraining.

### C.1.5   Identification task: M3CV [25]

The M3CV dataset[6] is a reliable brainprint identification system designed to withstand changes in the mental state of the subjects (cross-paradigm test) and successfully identify individuals even after several days (cross-session test). The Multi-Session Multi-Paradigm EEG database (Multi-Subject Multi-Session Multi-Paradigm Commonality and Variability (M3CV)) contains 106 healthy subjects, two sessions, and six types of EEG paradigms.

**Experimental paradigms:** Resting state, event-related potentials, evoked stimuli, P300, motor execution, and SSVEP.

**Preprocessing:** The full task-state data were used in the pretraining process of this paper, first upsampled to 256 Hz, and then standardized to mV, using a globally averaged reference, intercepting data from 0 to 4s after the start of each trial for pretraining.

---

[4]http://bci.med.tsinghua.edu.cn/download.html

[5]https://bcmi.sjtu.edu.cn/home/seed/

[6]https://aistudio.baidu.com/competition/detail/315/0/related-material

## C.2 DOWNSTREAM DATASET DESCRIPTION

### C.2.1 Motor Imagery Task: BCIC-2A [39]

This dataset[7] consists of EEG data from 10 subjects and includes four different motor imagery tasks: motor imagery of the left hand (Class 1), right hand (Class 2), feet (Class 3), and tongue (Class 4). Each subject performed two sessions on different days, with a total of 288 trials per session.

**Preprocessing:** Uniform units; 0 to 38 Hz filtering was used; resampling to 256 Hz; EA normalization [49] was used for each session. EA normalization is a common normalization method applied to data from motion imagery tasks such as BCIC2A.

**Experimental Configuration:** Experiments using the LOSO method. In the BCIC2A dataset 4-category cross-subject experiment, the accuracy of the model in this paper is lower than that of the task-specific SOTA model. This may be related to the high individual variability of the BCIC2A data. The fine-tuning method of combining the task-specific model with the model in this paper was used for testing. The MSConvNet model, which has a multi-scale convolutional EEGNet-like structure, was used as the motor imagery task-specific model. The model branch first maps the input 22-channel EEG signal into 10-20 system standard channel data using an adaptive spatial filter (convolution); then a pretrained encoder is used to extract the generic representations; and after collocating the generic representations and the in-domain representations of Wenchao's model branch, they are mapped into category logits using a linear layer. Training is performed using the AdamW optimizer, OneCycle learning rate strategy [43] (starting learning rate 5e-4, maximum 1e-3, minimum 4.22e-7), 100 rounds of training, and batch size of 72.

### C.2.2 Motor Imagery Task: BCIC-2B [40]

This dataset[8] consists of EEG data from 10 subjects. Each subject was given five training opportunities, the first two of which were training data without feedback (screening) and the last three with feedback. The three bipolar recordings (C3, Cz, and C4) were sampled at a frequency of 250 Hz. The dataset consisted of motor imagery (MI) for two categories: left-handed (category 1) and right-handed (category 2). Each subject participated in two sessions without feedback recordings on two different days within a two-week period. Each session consisted of 120 category-balanced trials.

**Preprocessing:** Uniform units; 0 to 38 Hz filtering was used; resampling to 256 Hz; EA normalization [49] was used for each session. EA normalization is a common normalization method applied to data from motion imagery tasks such as BCIC2A.

**Experimental Configuration:** Experiment using LOSO validation method. The fine-tuned modeling approach was used to test a cross-subject 2-categorization task on the BCIC2B dataset for the motor imagery task. The input data are 3-channel (C3, Cz, and C4 channels) EEG data with 4s, 256Hz sampling rate; to fully utilize the feature extraction capability of the model, spatial filters (convolution) were used to map the 3-channels of the input data into 7-channels, which corresponds to the C5, C3, C1, CZ, C2, C4, and C6 channels of the pretrained encoder; the features are extracted using the pretrained encoder and then mapped into 7 channels using the linear layer mapped to category logits. The training process is performed using AdamW optimizer, OneCycle learning rate strategy [43] (starting learning rate 1.6e-5, maximum 4e-4, minimum 1.51e-7), 100 rounds of training, and batch size of 64.

### C.2.3 Sleep Stage Detection Task: Sleep-EDFx [11]

This dataset[9] gives further insight into the generalizability of the model, as BCI data are typically categorized in the context of a specific trial or event, whereas SSC is a much more continuous problem that requires labeling the specific sleep stage that a subject is in over a long period of time. The Sleep-EDFx dataset contains 197 (78 healthy subjects) all-night sleep recordings, which include EEG, electrooculogram, chin EMG, and event markers.

**Preprocessing:** The preprocessing method of Banville et al. (2020) was referred to, which first converts the data unit to mV, and then uses a 30Hz low-pass filter, intercepts the samples in 30s

---

[7] https://www.bbci.de/competition/iv/#datasets
[8] https://www.bbci.de/competition/iv/#datasets
[9] https://physionet.org/content/sleep-edfx/1.0.0/

non-overlapping windows, and uses a channel-wise (channel-independent) z-score normalization for each sample.

**Experimental Configuration:** Subjects were randomly divided according to the ratio of 60

### C.2.4   ERN task: KaggleERN [41]

In this dataset[10], each subject is presented with letters and numbers (showing 36 possible items on a matrix) to spell words. Each item of the word is flashed in a random order grouped together and selected one at a time. The selected item is the one that the online algorithm is most likely to recognize as a typical target response. The goal of this challenge was to determine whether the selected items were correct by analyzing the brain signals of subjects after receiving feedback. Two transcription-spelling conditions were used: a fast mode with more errors (4 blinks per item); and a slower, less error-prone condition (8 blinks per item).

**Preprocessing:** Uniform units; downsampling to 256 Hz; global averaging reference; intercepting 2 s of data starting at -0.7 s before the onset of flicker.

**Experimental Configuration:** For comparison with the BENDR paper, 10 of the subject data were used as the test dataset, and the remaining subject data was used for 4-fold cross-validation method training. KaggleERN uses fine-tuning method 1, with a sliding window of 125ms in steps, to firstly map the 56 channels to the 19-channel EEG data of the 10-20 system by adaptive spatial filters (convolution), and then features are extracted using a pretrained encoder, and finally mapped into classification logits using a linear layer. The training process is performed using the AdamW optimizer, the OneCycle learning rate strategy [43] (starting learning rate of 1.6e-5, maximum of 4e-4, and minimum of 1.51e-7), and 100 rounds of training with a batch size of 64.

### C.2.5   P300 task: PhysioNetP300 [23]

In this dataset[11], each participant was asked to spell a total of 20 characters using a traditional matrix speller (Donchin speller). The target characters were randomly selected before the start of the run. Each row and column of a standard 6x6 character matrix was randomly augmented for 100 ms at 50 ms intervals with approximately 20 flashes. During this time, subjects were asked to focus their attention on the target character and mentally count the number of times the target character was highlighted.

**Preprocessing:** Uniform units; 120Hz low-pass filtering was used; downsampling to 256Hz; and 2s of data starting at -0.7s before the onset of the flicker was intercepted.

**Experimental Configuration:** For comparison with the BENDR paper, subjects 8, 10, and 12 were removed and the data from the remaining 9 subjects were retained. Experiments were performed using the LOSO cross-validation method. The PhysioNetP300 uses a fine-tuning method with a sliding window of 125ms step size, using all channels used in the pretraining phase; the data from each EEG channel is first adaptively scaled by an adaptive spatial filter (learnable scaling factor), then features are extracted using the pretraining encoder, and finally mapped using the linear layer mapping for classification logits. The training process uses AdamW optimizer, OneCycle learning rate strategy [43] (starting learning rate 3.2e-5, maximum 8e-4, minimum 3.02e-7), 100 rounds of training, and batch size of 64.

### C.2.6   TUAB (abnormal detection) and TUEV (event type classification) [18]

TUAB[12] is a corpus of EEGs which are 23-channel and sampled at 256 Hz. All data have been annotated as normal or abnormal. There are total 409,455 10-second samples that we use for binary classification to predict normal/abnormal.

TUEV[13] is a subset of TUEG that contains annotations of EEG segments as one of six classes: (1) spike and sharp wave (SPSW), (2) generalized periodic epileptiform discharges (GPED), (3) periodic lateralized epileptiform discharges (PLED), (4) eye movement (EYEM), (5) artifact (ARTF) and

---

[10]https://www.kaggle.com/c/inria-bci-challenge/data
[11]https://physionet.org/content/erpbci/1.0.0/
[12]https://isip.piconepress.com/projects/tuh_eeg/html/downloads.shtml
[13]https://isip.piconepress.com/projects/tuh_eeg/html/downloads.shtml

(6) background (BCKG). The EEG signals contain 23 channels at 256 Hz and are segmented into 112,491 5-second samples.

We conducted experiments using linear-probing on these two datasets. Since these datasets use different electrodes and sampling rates compared to the model pretraining, we used two layers of convolution (spatial convolution and temporal convolution) to adaptively fit the data. The spatial convolution used a 1x1 convolution, and the temporal convolution used depthwise convolution. The convolution kernel size for TUAB was (1, 15), and for TUEV, it was (1, 55). Both experiments used the same optimizer and a learning rate of 5e-4. Due to GPU memory limitations, the batch size for TUAB was 100, and for TUEV, it was 500.

Table 12: Model design for TUAB dataset.

| Input Size | Operator | kernel | stride | groups | padding |
|---|---|---|---|---|---|
| $23 \times 2000$ | conv1d | 1 | 1 | 1 | 0 |
| $20 \times 2000$ | batchnorm,gelu | - | - | - | - |
| $20 \times 2000$ | conv1d | 15 | 1 | 20 | 7 |
| $20 \times 2000$ | batchnorm,gelu | - | - | - | - |
| $20 \times 2000$ | dropout(0.25) | - | - | - | - |
| $20 \times 2000$ | eegpt-encoder | 64 | 64 | - | - |
| $31 \times 4 \times 512$ | flatten,linear | - | - | - | - |
| 1 | output | - | - | - | - |

Table 13: Model design for TUEV dataset.

| Input Size | Operator | kernel | stride | groups | padding |
|---|---|---|---|---|---|
| $23 \times 1000$ | conv1d | 1 | 1 | 1 | 0 |
| $20 \times 1000$ | batchnorm,gelu | - | - | - | - |
| $20 \times 1000$ | conv1d | 55 | 1 | 20 | 27 |
| $20 \times 1000$ | batchnorm,gelu | - | - | - | - |
| $20 \times 1000$ | dropout(0.5) | - | - | - | - |
| $20 \times 1000$ | eegpt-encoder | 64 | 64 | - | - |
| $15 \times 4 \times 512$ | flatten,linear | - | - | - | - |
| 6 | output | - | - | - | - |

The detailed modelling structure used in the downstream tasks is presented in Table 12 and Table 13. The 23-channel input is first to reduce the number of channels to 20 by the convolution. Then, the eegpt-encoder uses the following 20 channel embeddings as the inputs' channel embeddings: [FP1, FPZ, FP2, F7, F3, FZ, F4, F8, T7, C3, CZ, C4, T8, P7, P3, PZ, P4, P8, O1, O2]. The eegpt-encoder maps 64-length window segments of the input signals to 4 (number of summary tokens) × 512-dimensional features. Finally, the flatten and linear layers are used to output the final classification score.

# D   METRICS DESCRIPTION

The following metrics are used for comparison: 1) Balanced Accuracy (BAC): The mean recall across all classes, applicable to both binary and multi-class classification. 2) AUROC: The area under the receiver operating characteristic curve, primarily used for binary classification. 3) Weighted F1: The harmonic mean of precision and recall, with equal emphasis on both metrics, employed for evaluating multi-class classification. 4) Cohen's Kappa: A statistic measuring the agreement between categorical variables X and Y, derived from the observed and expected frequencies in the diagonal of a square contingency table. We set AUROC as the monitor score for binary classification and Cohen's Kappa as the monitor score for multi-class classification.

# E  MORE IMPLEMENTATION DESCRIPTION

For model implementation, the BENDR code is downloaded and modified upon the github[14], the BENDR code is downloaded and modified upon this github[15], and the LaBraM code is downloaded and modified upon this github[16]. Other baselines implementations on the TUAB and TUEV datasets are also referred to this repo[17]. The EEGPT structure implementation is mainly modified upon the github[18], and also references the code of BENDR. The ROPE embedding is implemented using the code of Roformer [50] in github[19].

We mainly use MNE package[20], braindecode package[21], and torcheeg package[22] to load and preprocess the data.

# F  MORE EXPERIMENT RESULTS DESCRIPTION

In the comparative experiments, the baseline model BIOT demonstrated an accuracy of 54% in the binary classification task of EEG P300 signals on PhysioP300 dataset4, suggesting its ineffectiveness.

The P300 signal is an event-related potential (ERP) component commonly observed in EEG recordings. It is characterized by a positive deflection in voltage occurring approximately 300 milliseconds after the presentation of a stimulus. The P300 component is primarily concentrated around the 300 ms mark and is crucial for tasks involving attention and stimulus evaluation.

One plausible reason for the suboptimal performance of the BIOT model is the inherent nature of the P300 signal classification task, which predominantly relies on time-domain waveform features. The BIOT model, however, applies a fast Fourier transform (FFT) to the input signals, thereby primarily extracting frequency-domain features. This fundamental discrepancy between the feature domain utilized by the model and the domain that is most pertinent to the classification task likely accounts for the observed lack of efficacy. Additionally, while the P300 component is concentrated around the 300 ms mark, the BIOT model applies an FFT on 1-second length patches, which is overly extensive and results in significant information loss. The "significant information loss" is mainly reflected in the fact that the BIOT model only retains the spectral energy information for 1s of each patch after the FFT, and discards the phase information. A similar situation also occurs in the binary classification task for ERN signals [27] on the KaggleERN dataset.

# G  VISUALIZATION

## G.1  Channel Relationship Visualization

To further demonstrate the effectiveness of the model implementation, this paper visualizes the trained model. During the pretraining process, channel information is encoded through learnable channel embeddings. This section visualizes the similarity between positional encodings to show the relationship information learned by the model from the data.

Figure 11(a) shows the channel relationship diagram after model pretraining, using the cosine similarity between channel embeddings to represent the relationships between channels. The left figure shows relationships with similarity greater than 0.5, where channels are clustered based on their front, back, left, and right positions. The right figure shows relationships with similarity greater than 0.4, revealing not only nearby relationships but also long-distance relationships across brain regions.

---

[14]https://github.com/SPOClab-ca/BENDR
[15]https://github.com/ycq091044/BIOT
[16]https://github.com/935963004/LaBraM
[17]https://github.com/ycq091044/BIOT
[18]https://github.com/google-research/vision_transformer
[19]https://github.com/ZhuiyiTechnology/roformer
[20]https://github.com/mne-tools/mne-python
[21]https://github.com/braindecode/braindecode
[22]https://github.com/torcheeg/torcheeg

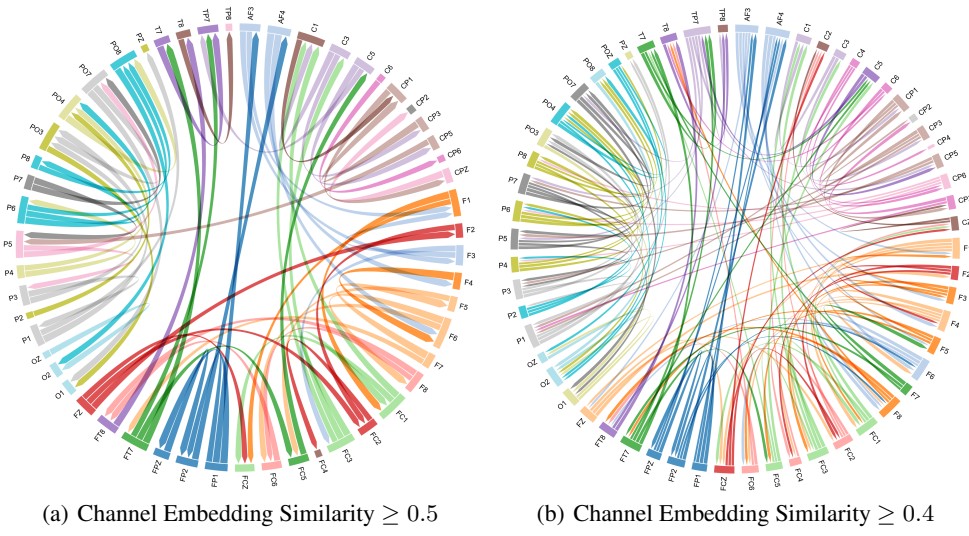

(a) Channel Embedding Similarity $\geq 0.5$      (b) Channel Embedding Similarity $\geq 0.4$

Figure 11: Channel Embedding Similarity Relationships

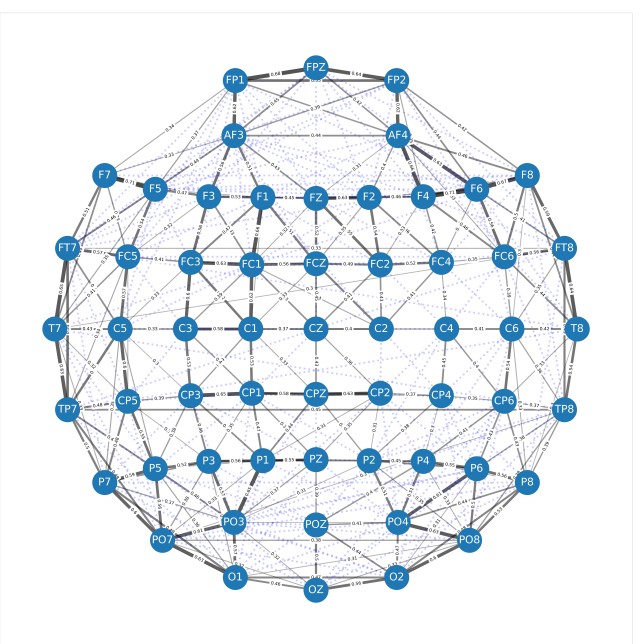

Figure 12: Channel Embedding Similarity Connection Diagram

Figure 12 shows the cosine similarity relationships between channel embeddings, with electrode positions placed according to their actual locations. Solid lines indicate relationships with similarity greater than 0.3, while dashed lines indicate relationships with similarity between 0.1 and 0.3. It can be seen that channels in close proximity have higher similarity, and there is also significant similarity between distant electrodes on opposite sides.

## G.2 BCIC2A Experiment Results Visualization

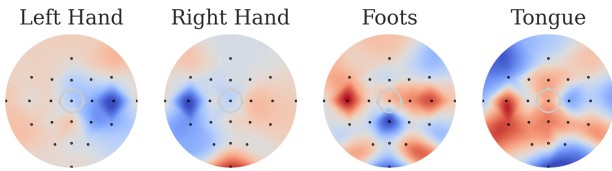

Figure 13: BCIC2A Channel and Class Pearson Correlation Diagram

Figure 13 shows the correlation between channels and motor imagery classes detected using the channel perturbation method after training on the BCIC2A dataset. Gaussian multiplicative random noise is randomly added to the signal amplitude of each channel, and the Pearson correlation between the noise intensity and the changes in the corresponding class logits is calculated and presented as a heatmap. Symmetric relationships are observed for electrodes related to left and right hand movements, with bilateral electrodes corresponding to foot movements, and distinct channels corresponding to the four different classes.

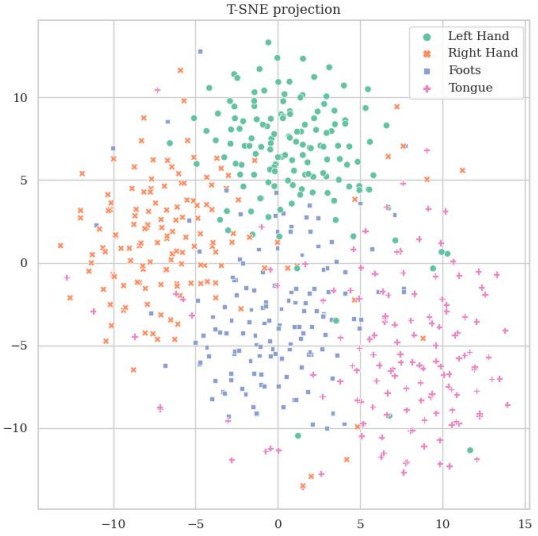

Figure 14: BCIC2A t-SNE Diagram

Figure 14 shows the t-SNE diagram of features learned by the model. It can be seen that the features of the four classes are clustered in four different regions, demonstrating linear separability.

Figure 15 shows the confusion matrix, indicating that the recognition performance for foot movements is the best, while samples from other classes are often misclassified as foot movements.

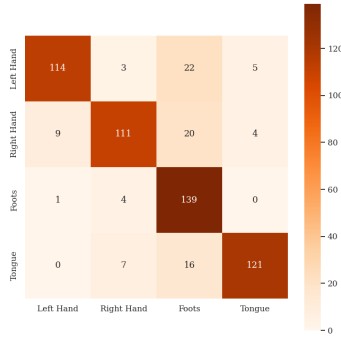

Figure 15: BCIC2A Confusion Matrix Diagram

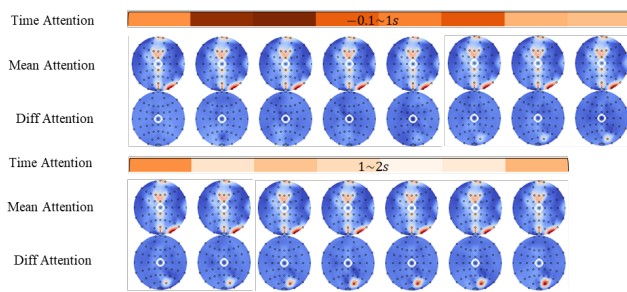

Figure 16: P300 spatio-temporal Attention Diagram

## G.3 PhysioP300 Experiment Results Visualization

Figure 16 shows the model's attention distribution for the P300 task. For temporal attention, the model focuses more on the -0.1 to 1s time period. For spatial attention, the model focuses more on the signals from the F1, FZ, F2, and FCZ channels.

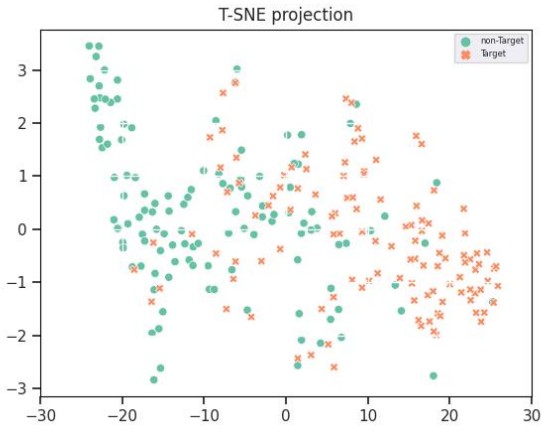

Figure 17: P300 t-SNE Diagram

Figure 17 shows the t-SNE diagram of features extracted by the model for the two classes of samples. The features of the two classes are distributed on two sides, with a confusion belt in the middle, but overall the classes are linearly separable.

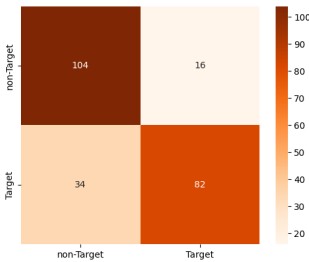

Figure 18: P300 Confusion Matrix Diagram

The confusion matrix shows that the performance for the non-target class is better.

# H   LIMITATIONS

Firstly, although we have collected a large-scale multi-task mixed EEG dataset and utilized a model structure with extensive parameters during the pretraining phase, there remains a significant disparity compared to today's large vision and language models. Our work is still in the exploratory phase of training large EEG models to learn general representations. Encouragingly, our experiments have shown that large EEG models can continue to be optimized, achieving considerable performance improvements compared to existing methods developed for specific BCI tasks and general large EEG models. Secondly, while EEGPT requires only linear-probing to adapt to small-scale downstream tasks, it may still result in high memory costs. Finally, EEGPT was pretrained using 4s EEG data. This may strict model's capability. It is worth exploring the training of large EEG models with long recordings.

