# OpenReview forum: "EEGPT: Pretrained Transformer for Universal and Reliable Representation of EEG Signals"
_NeurIPS.cc/2024/Conference — NeurIPS 2024 poster_

### Official Review · Reviewer_izki · 2024-07-01

**Soundness:** 3
**Presentation:** 3
**Contribution:** 2
**Rating:** 5
**Confidence:** 4

**Summary:**

The paper presents EEGPT, a 10-million-parameter pretrained transformer model designed for universal EEG feature extraction. The model employs a dual self-supervised learning method for efficient feature extraction and demonstrates state-of-the-art performance on various downstream tasks with linear probing.

**Strengths:**

1. The proposed EEGPT model introduces a dual self-supervised learning method combining spatio-temporal representation alignment and mask-based reconstruction, which enhances representation quality and model robustness.
2. EEGPT achieves state-of-the-art performance on a range of downstream tasks such as motor imagery classification, ERP detection, and sleep stage detection.
3. The hierarchical structure for decoupled processing of spatial and temporal information reduces computational complexity and enhances flexibility for BCI applications.
4. Comprehensive experiments demonstrate the superior performance of EEGPT across multiple EEG tasks.

**Weaknesses:**

1. The function of the adaptive spatial filter is unclear. Why is this module not included during pre-training? An ablation study is needed to evaluate the effectiveness of this component.
2. In Table 2, the authors use different metrics than in Table 3. Cohen's Kappa should be used consistently across both tables. Additionally, the results of LaBraM on these two datasets should be reported to provide a more comprehensive comparison, even if EEGPT, which uses linear probing, might not surpass LaBraM.
3. The technical contribution of the paper appears incremental. The framework is largely based on the context autoencoder (CAE) with modifications such as the adaptive spatial filter, multiple summary tokens, and rotary temporal embeddings.
4. During pre-training, EEGPT uses EEG data of the same configuration. In contrast, LaBraM leverages various datasets with different configurations, which might make EEGPT less universal.
5. The ablation study is conducted on a single dataset, which is not persuasive enough to demonstrate the effectiveness of the proposed modules and their scalability.
6. EEGPT uses different model settings for different downstream datasets, which may limit its generalizability.

**Questions:**

1. It is unclear whether the baselines in Table 3 are based on linear probing or fully fine-tuned models. Clarification is needed.

**Limitations:**

The authors have discussed the limitations.

---

> ### Author Rebuttal · Authors · 2024-08-05
>
> We thank the reviewer for the helpful feedback. We have uploaded a revision with the changes marked as blue. Our detailed responses are as follows:
>
> ---
>
> #### **W1: The function of the adaptive spatial filter is unclear. Why is this module not included during  pretraining? An ablation study is needed to evaluate the effectiveness of this component.**
>
> The adaptive spatial filter is designed for the downstream task, adapting the EEG signal for each task/data distribution to the input of the encoder. In the downstream task, the model learns to adjust the parameter weights of the adaptive spatial filter, which maps (scale and project) the downstream task data to the same distributions as the pretrain data. For example, it can adapt to the different EEG references (e.g., for the epilepsy classification task, the double banana bipolar montage reference [2] with the features are more significant, and adaptive spatial filters allow for such a reference.).
>
> The adaptive spatial filter is designed only for downstream tasks. It can be of limited use during  pretraining, since we have uniformly pre-processed the  pretraining dataset and assumed that these data are in the same distribution. On the other hand, during  pretraining, since the strategy of masking in both the spatial and temporal dimensions is used, the channels are not consistent in each input, and consequently leads to distributional drift and reduces training efficiency [3].
>
> In the ablation experiments, we added the results of the ablation experiments that exclude the adaptive spatial filter in the downstream task, as shown in Revision Appendix A.3 Table 8 & author rebuttal Table 2.
>
> #### **W2: In Table 2, the authors use different metrics than in Table 3. Cohen's Kappa should be used consistently across both tables. Additionally, the results of LaBraM on these two datasets should be reported to provide a more comprehensive comparison, even if EEGPT, which uses linear probing, might not surpass LaBraM.**
>
> We added the Cohen's Kappa metrics to Table 3 and added the LaBraM-Base results, see Revision Appendix A.6.
>
> **[Table: Results of experiments on TUAB]**
>
> |Methods|Model Size|Balanced Accuracy|AUROC|
> |-|-|-|-|
> |BIOT|3.2M|0.7959±0.0057|0.8815±0.0043|
> |LaBraM|5.8M|0.8140±0.0019|0.9022±0.0009|
> |Ours-Tiny|4.7M|0.7959±0.0021|0.8716±0.0041|
> |Ours|25M|0.7983±0.0030|0.8718±0.0050|
>
> **[Table: Results of experiments on TUEV]**
>
> |Methods|Model Size|Balanced Accuracy|Weighted F1|Cohen's Kappa|
> |-|-|-|-|-|
> |BIOT|3.2M|0.5281±0.0225|0.7492±0.0082|0.5273±0.0249|
> |LaBraM|5.8M|0.6409±0.0065|0.8312±0.0052|0.6637±0.0093|
> |Ours-Tiny|4.7M|0.5670±0.0066|0.7535±0.0097|0.5085±0.0173|
> |Ours|25M|0.6232±0.0114|0.8187±0.0063|0.6351±0.0134|
>
> LaBraM employs a larger  pretraining dataset than ours (reference to [1] Appendix D), which also contains the TUEG dataset that is similar to the TUAB and TUEV distributions, and whose paper illustrates the scale effect of the amount of  pretraining data, which may have facilitated the learning of a richer and more downstream task-adaptable representation for LaBraM.
>
> #### **W4: During  pretraining, EEGPT uses EEG data of the same configuration. In contrast, LaBraM leverages various datasets with different configurations, which might make EEGPT less universal.**
>
> In this paper, we unified the  pretraining dataset into 58 channels, each channel has the same amount of data, making each channel of equal importance. In addition to this, LaBraM uses more eeg reference configuration (the double banana bipolar montage reference [2]) for epilepsy data, which may lead to stronger generalization (for epilepsy tasks). We conducted initial experiments using a limited dataset and will gradually expand to larger datasets (e.g., epilepsy dataset) to enhance the generalizability of EEGPT.
>
> #### **W5: The ablation study is conducted on a single dataset, which is not persuasive enough to demonstrate the effectiveness of the proposed modules and their scalability.**
>
> In the ablation experiment, we added test results for the BCIC-2B and KaggleERN datasets, as shown in the table below, and see Revision Section 3.4. See author rebuttal Table 1 for more details.
>
> #### **Q1: It is unclear whether the baselines in Table 3 are based on linear probing or fully fine-tuned models. Clarification is needed.**
>
> The baseline in Table 3 uses the full fine-tuning method, which we have supplemented, see Revision Section 3.2.
>
> [1] Jiang, W. B., Zhao, L. M., & Lu, B. L. (2024). Large brain model for learning generic representations with tremendous EEG data in BCI. *arXiv preprint arXiv:2405.18765*.
>
> [2] Rosenzweig, I., Fogarasi, A., Johnsen, B., Alving, J., Fabricius, M. E., Scherg, M., ... & Beniczky, S. (2014). Beyond the double banana: improved recognition of temporal lobe seizures in long-term EEG. *Journal of Clinical Neurophysiology*, *31*(1), 1-9.
>
> [3] Tian, K., Jiang, Y., Diao, Q., Lin, C., Wang, L., & Yuan, Z. (2023). Designing bert for convolutional networks: Sparse and hierarchical masked modeling. arXiv preprint arXiv:2301.03580.
>
> ---
>
> Thanks again for your comments. Hope our rebuttal has addressed all your concerns.

---

> > ### Comment · Reviewer_izki · 2024-08-12
> >
> > I appreciate the comments by the authors. Some of my concerns are addressed by the clarification and additional experiments. However, regarding the technical contribution and universibility (W3 and W4), I maintain my standpoints.

---

### Official Review · Reviewer_23GR · 2024-07-08

**Soundness:** 2
**Presentation:** 3
**Contribution:** 3
**Rating:** 6
**Confidence:** 4

**Summary:**

The authors propose a novel pretraining strategy, EEGPT, that is essentially a multi-task self-supervision loss consisting of a masked autoencoder-style reconstruction objective, and an alignment loss that is reminiscent of knowledge distillation approaches such as data2vec. EEGPT is applied to representation learning of electroencephalography (EEG) data, and the authors demonstrate the pretrained model flexibility across a variety of downstream EEG decoding tasks (motor imagery, sleep staging, ERP detection).

**Strengths:**

a. Originality:  The individual components of EEGPT are not original. The masking strategy and reconstruction loss are based on Masked Autoencoder which was originally developed for vision, and has since been applied to time-series data (including EEG) previously. Similarly, the alignment loss is highly similar to that of data2vec, except that data2vec uses a smoothed L1 loss. However, the combination of these two objectives into a multi-task self-supervision loss is novel (to my knowledge). According to the ablation results in Table 5, it does seem that the two objectives compete against each other, but in a way that is constructive for the downstream decoding tasks.

b. Quality: The conclusions made by this paper are mostly sound. The authors demonstrate the improvements of EEGPT over other SOTA approaches across a comprehensive list of downstream EEG decoding tasks. Also of note, the authors made an effort to demonstrate the design choices of EEGPT in their ablation analysis.

c. Clarity: The paper is overall clear in its writing. There are some missing details, which I have detailed below in the Questions section.

d. Significance: The pretraining approach is likely to be reused in other works. The model generalizability across a diverse set of EEG decoding tasks is especially compelling, since it suggests that this model may be used as a generalized foundation model for EEG.

**Weaknesses:**

b. Quality: One weakness of this work is that the proposed model has significantly more parameters than the SOTA approaches it is comparing against. It remains unclear if the authors have developed a better pretraining approach, or if the improvements in the downstream tasks are due to higher expressivity in the base model architecture. Could the authors add additional comparisons where the number of parameters EEGPT matches the number of parameters in the other SOTA approaches?

Another weakness is that the comparison to pretraining approaches such as BIOT is unfair. The authors state in Appendix E: “the BIOT model applies an FFT on 1-second length patches, which is overly extensive and results in significant information loss". However, this does not mean that the pretraining recipe in this paper is better than BIOT. It could be the case that BIOT performs similarly, it just needs to be retrained with different patch length and/or patch stride. I understand that retraining BIOT with new parameters may not feasible in the rebuttal timeframe, but I think this should be clarified in the text.

**Questions:**

1. Line 59: The authors claim that the low SNR of EEG “...make it challenging to learn abstract features using masked autoencoders”. However, there have been multiple works that use MAE-style pretraining to successfully learn features from EEG, such as: [1] Chien, et al. MAEEG: Masked Autoencoder for EEG Representation Learning. arXiv:2211.02625. [2] Wu, et al. Neuro-BERT: Rethinking Masked Autoencoding for Self-supervised Neurological Pretraining. IEEE JBHI.
2. Eqn 9: Did the authors try using a weighted loss with weight $\lambda$? Or does each individual loss take on similar values? $\mathcal{L} = \mathcal{L}_A + \lambda \mathcal{L}_R$
3. Section 2.4: Can the authors clarify which “Features” are passed to the linear classification head? Are these the summary tokens (passed to the Predictor during pretraining)? Or are they the non-summary tokens?
4. Section 3.1: Were any notch filters used to suppress electrical noise in datasets used for pretraining or downstream tasks?
5. Section 3.2: Why were 58 electrodes used? Is it because these 58 electrodes were common across the pretraining datasets? Also, what happens at inference time if a new electrode not belonging to the original set of 58 locations appears?
6. Section 3.2: Is there any normalization applied to the input signal? I saw in the appendix that each dataset has a different normalization (EA on entire session, z-score on each input, etc.). Does that mean that the normalization is dataset-dependent?
7. Line 197: Missing reference for OneCycle learning rate
8. Tables 2&3: The proposed EEGPT model has ~8X as many parameters as the second largest model (BIOT). It would be a fairer comparison if the authors also reported BAC for a similarly sized EEGPT model, such as tiny3 or little (from Table 6).
9. Table 5: Can more description be added to this table? For example, BAC stands for Balanced Accuracy I assume? And which dataset is BAC being reported for here? Or is it an average across datasets?
10. (Minor comment) The title “EEGPT” is slightly misleading, since it makes the reader think that the model is being pretrained using an autoregressive reconstruction objective such as in the popular GPT language models (e.g. predict the next token given the previous tokens). I would recommend changing to a different name to avoid confusion.
11. (Minor comment) Line 186: Spelling error “Appendix”
12. (Minor comment) Line 499: What does EA in “EA normalization” stand for? I tried to find it elsewhere in the paper, but could not find it..

**Limitations:**

Yes.

---

> ### Author Rebuttal · Authors · 2024-08-05
>
> We thank the reviewer for your appreciation and constructive comments. We have uploaded a revision and used blue to mark the new changes.
>
> ---
>
> #### **W1&Q8**
>
> See Revision Section 3.3 for test results of model variants with comparable number of parameters.
>
> **[Table: Results of experiments on TUAB]**
>
> |Methods|Model Size|Balanced Accuracy|AUROC|
> |-|-|-|-|
> |BIOT|3.2M|0.7959±0.0057|0.8815±0.0043|
> |Ours-Tiny|4.7M|0.7959±0.0021|0.8716±0.0041|
> |Ours|25M|0.7983±0.0030|0.8718±0.0050|
>
> **[Table: Results of experiments on TUEV]**
>
> |Methods|Model Size|Balanced Accuracy|Weighted F1|Cohen's Kappa|
> |-|-|-|-|-|
> |BIOT|3.2M|0.5281±0.0225|0.7492±0.0082|0.5273±0.0249|
> |Ours-Tiny|4.7M|0.5670±0.0066|0.7535±0.0097|0.5085±0.0173|
> |Ours|25M|0.6232±0.0114|0.8187±0.0063|0.6351±0.0134|
>
> The number of parameters in the table for Ours-Tiny is 4.7M comparable to the BIOT model. Compared to BIOT, our Tiny model improves about 4% on the TUEV dataset and achieves comparable performance with it on the TUAB dataset. Compared to the BIOT method, we used a smaller pretraining dataset and did not use more datasets containing epilepsy samples, but still achieved good results on both datasets.
>
> #### **W2**
>
> We agree with you that on some tasks it is possible that the pretraining scheme in this paper performs similarly to the BIOT. The statement in Revision Appendix F aims to explain the possible reasons for the poor performance of the BIOT on the BCIC-2A and BCIC-2B motor imagery tasks (see Section 3.3, Table 4). The "significant information loss" is mainly reflected in the fact that the BIOT model only retains the spectral energy information for 1s of each patch after the FFT, and discards the phase information. We have provided additional explanations for this, see Revision Appendix F.
>
> #### **Q1**
>
> While other work has validated the ability of MAE pretraining methods to learn features in corresponding EEG tasks, our work would like to emphasize that the low signal-to-noise ratio of EEG prevents the model from learning "high-quality" representations through MAE pretraining methods. Our ablation experiments verify this, see Revision Section 3.4.
> See author rebuttal Table 1 for more details. In the absence of $L_A$ loss, the model's performance degradation on all datasets is significant (6% to 9%).
>
> #### **Q2**
>
> In our experiments, despite the presence of two loss functions, no weights were explicitly introduced during the optimization process to balance the two losses. This approach also simplifies the training process of the model.
>
> #### **Q3**
>
> "The summary tokens (passed to the Predictor during pretraining)" are passed to the linear classification head. We provide additional clarification, see Revision Section 2.4.
>
> #### **Q4**
>
> In Revision Appendix C we describe in detail how we preprocessed each dataset. For all datasets we did not target the use of notch filters to suppress electrical noise.
>
> #### **Q5**
>
> This is because these 58 electrodes are the electrodes that are present in all pretraining datasets and are the intersection of their electrode sets. These 58 electrodes cover as much as possible the electrodes in the international 10-10 EEG system standard [1]. During the pretraining phase, we removed channel data from the data sample that were not these 58 electrodes.
>
> If a new electrode that does not belong to the 58 pretrained electrodes appears in the downstream task, the signal from the new electrode can be mapped to the neighbouring/similar electrode input of the encoder model by using the adaptive spatial filter (see Revision Section 2.4), which is done by training the adaptive spatial filter on the downstream task dataset.
>
> #### **Q6**
>
> In Revision Appendix B, we describe in detail the normalization and preprocessing approach for each dataset. The normalization and preprocessing approach is dependent on different EEG paradigms.
>
> #### **Q7**
>
> We have added references to OneCycle Learning Rate [2], see Revision Section 3.2 & Appendix C.2.
>
> #### **Q9**
>
> We added more detailed descriptions for the tables, and in Table 5 of Revision Section 3.4, we changed the headers to BCIC-2A-BAC, BCIC-2B-AUROC, and KaggleERN-AUROC.
>
> #### **Q10**
>
> Thanks for commenting on the potential confusion with "EEGPT". We chose this to highlight the application of pretrained Transformers on EEG signals, despite not using autoregressive task pretraining. It showcases the ability of to learn universal pattern from massive data. However, we've clarified that our model isn't pretrained using autoregressive objectives in our paper's introduction (Revision Section 1) to prevent misunderstandings, while maintaining our original title intent.
>
> #### **Q11**
>
> We have fixed spelling errors, see Revision PDF.
>
> #### **Q12**
>
> We have added a citation and description of EA normalisation [3], see Revision Appendix C.2. EA normalization is a common normalization method applied to EEG data from motor imagery tasks. This approach align EEG trials from different subjects in the Euclidean space to make them more similar, and hence improve the learning performance for a new subject.
>
> [1] The five percent electrode system for high-resolution EEG and ERP measurements.
>
> [2] Super-convergence: Very fast training of neural networks using large learning rates.
>
> [3] Transfer learning for brain–computer interfaces: A Euclidean space data alignment approach.

---

> > ### Comment · Reviewer_23GR · 2024-08-10
> >
> > Thank you for thoroughly addressing my comments & questions. See below for further comments
> >
> > New comment: I see that the authors have added a comparison to LaBraM in A.6 showing that LaBraM outperforms EEGPT significantly on TUEV and TUAB datasets. However, Table 4 demonstrates that EEGPT is better than LaBraM at other tasks (BCIC, SleepEDF, etc.). Do the authors have any explanation for this?
> >
> > W1: Thanks for running this! Indeed, it seems like when the number of parameters is matched between EEGPT and BIOT the performance is better (TUEV) or similar (TUAB).
> >
> > W2: I appreciate that you added the comment about possible reasons for why BIOT might be worse in the Appendix. I still feel the comparison is slightly unfair, and re-running the BIOT pretraining with smaller token sizes should be subject of future work to determine whether EEGPT is truly a better pretraining strategy.
> >
> > Q12: Can you define the acronym for EA somewhere in the text? I assume it stands for Euclidean Alignment?

---

> > > ### Author Response · Authors · 2024-08-11
> > >
> > > #### **New comment**
> > > Our dual self-supervised method improves the quality of EEG representations, therefore it performs better on other tasks.
> > >
> > > On the TUAB and TUEV tasks, LaBraM may achieve better performance for two reasons: firstly, LaBraM uses 8-second pre-training data segments (while EEGPT uses 4 seconds), and secondly, LaBraM employs TUEG datasets (which are similar to the data distribution of TUAB and TUEV) as pretraining datasets, thus it is able to better capture the long-term features in the EEG data.
> > >
> > > We use a 4-second pretraining dataset because all the pre-training data we use are task-state data segments, which are almost 4 seconds in length to ensure data quality.
> > >
> > > #### **W2**
> > > In this paper, our work mainly focuses on showing that, under the same conditions, our proposed dual self-supervised method demonstrates significant performance improvement compared to the MAE self-supervised method. Moreover, our model exhibits comparable or better performance on a wider range of tasks/datasets compared to other SOTA general EEG pretrained models (not pretraining methods), suggesting that our model may have stronger generalizability.
> > >
> > > The main improvement of BIOT lies in the Biosignal Tokenization method, which is an improvement in the tokenizer, such as the introduction of normalization and FFT to retain more task-related features; while LaBraM's main improvement is the proposal of the Neural Tokenizer inspired by VQ-VAE, and using the tokens encoded by the Neural Tokenizer as the prediction target, which is an improvement in the prediction target, making the prediction target contain more task-related features (such as spectral features); in contrast, our work focuses on an enhancement to the self-supervised method itself, using a dual self-supervised method to enhance the quality of the representations.
> > >
> > > EEGPT, BIOT, and LaBraM may each have their own design advantages in the pretraining method. More work may be needed in the future for a more fair comparison, and they can learn from and integrate with each other.
> > >
> > > #### **Q12**
> > > Thank you for your comments. EA stands for 'Euclidean space EEG data alignment' approach, which we have declared in the new revision.

---

### Official Review · Reviewer_DgXb · 2024-07-10

**Soundness:** 3
**Presentation:** 2
**Contribution:** 2
**Rating:** 5
**Confidence:** 4

**Summary:**

The paper presents a new pretrained model called EEGPT (EEG pretrained transformer), designed to improve the analysis of EEG (electroencephalography) signals. EEGPT is a model with over 10 million parameters (up to 100M for the largest model) that aims to solve common problems in EEG analysis, where the challenges are low signal quality and high variability between individuals. The model introduces a special method called "dual self-supervised learning" to extract better features from EEG data. This method includes two main techniques: the usual masked reconstruction loss, and additionally the alignment loss. The alignment loss is a special loss introduced inside the masked auto-encoder to “force” the learned features of the masked patches inside the encoder (and predictor) to be aligned with spatial and temporal patches (all masked and unmasked). This is achieved by adding a momentum encoder and computing the loss between the predictor (global integration of summary patches of each time j) and momentum encoder (spatial integration of all unmasked patches for each time j).

The model was trained on several large EEG datasets and tested on various tasks like classifying motor imagery, detecting event-related potentials, and identifying sleep stages. EEGPT performed better than existing models in these tasks, showing it is effective and scalable.

**Strengths:**

1. The model introduces the “dual self-supervised technique” and to my knowledge it is novel.
2. The model performs better in the proposed tasks presented in the paper. However, the model is also significantly larger than the competition.
3. The paper performs an ablation study showing the importance of their decision of using the alignment loss (L_A). Without L_A, accuracy drops from 58.46 to 52.87.
4. The paper presents a range of experiments besides the model training and evaluation, providing a better understanding of the model design and behaviour. This includes methods like scaling laws, relationships of the channel embeddings (which are learned), and visualizations on the correlations of the motor imagery tasks and the channels for the BCI competition (BCIC2A dataset).
5. The downstream tasks are performed only with linear probing (no need for full finetuning) and the model still outperforms other models.

**Weaknesses:**

1. The experiments done for the model design and behaviour such as scaling laws, ablation studies, channel embedding relationships, etc, are done only on one downstream task: namely the BCIC2A dataset. How does the model perform on the other tasks, what is the reason for not presenting the results on the other datasets?
2. The paper presents interesting scaling law experiments, showing potential in the method to scale to larger model size. However the experiments are too limited to be able to state the scaling laws as given in the paper (e.g. the ACC = (33.6*N)^0.029). I would argue one has to test it either to more tasks, or have a unified metric that uses many datasets for the scaling law.  Also, how does the law depend on the amount of training data used?
3. The design of the model was at first confusing and I still have trouble understanding the main motivation from some of the decisions. The ablation studies are a great help here, but I think a few more ablation studies or explanations on the design of the model would clarify the paper better (see Questions section).
4. I have a few doubts on the evaluation of the baselines. Although I am not certain what are the best achievable accuracies in many of the tasks, for TUAB we have many different models that might perform better: We have models up to 86% (ChronoNet) in Table 1 in here (https://www.sciencedirect.com/science/article/pii/S2213158223001730) and above 90% in here (https://arxiv.org/pdf/2401.10283). I know there are certain differences: accuracy vs balanced accuracy and the second paper has a different evaluation for the highest accuracy. However, the TUAB dataset is quite balanced to my knowledge and still in the second paper in Table 1, there are multiple models close to 90%. Why are these models not included in the comparison, and since we have this discrepancy in this task, do other models exist also for other tasks as well?
5. Why is the pretraining set very small? In terms of participants, the downstream set is much bigger than the pretraining set. In addition, the TUEG dataset is bigger than all the datasets used here (combined). Is there any reason why the authors chose these datasets in particular? It is not clear from the paper why this is the case. Usually, during pretraining the dataset is much bigger compared to the downstream task. In this paper this does not seem to be the case. In fact, many of these datasets (all?) used during pretraining do have labels (i.e. why self-supervised).
6. In some sections it was not clear which evaluation dataset has been used to produce the results. For example in Section 3.4 there is Table 5 but it is not clear which model size and which dataset has been used to produce the results. Because of the accuracy number 58.46 I could infer that it was BCIC2A, but it is not mentioned anywhere. For scaling laws it is mentioned in the text, but it is better to have it on the table description.
7. Many minor annoyances, which make me wonder how much care the authors took to write this paper:
-  Misspellings, for example: Accuarcy vs accuracy, Appandix vs Appendix, etc.
- The references are messed up. Just look at the first two references [1] and [2], where every author is listed at least 2 times, sometimes even 3 times!
- You need to put a space in front of a reference, like this [1]. You almost always have no space, but not even this is consistent, because sometimes you do.

**Questions:**

Even though there is the ablation study for the alignment loss, it is not clear from the paper what is the main motivation behind it, and why this particular design is used. To be more concrete, we have the summary tokens from the encoder which are only created from patches at specific time point j. By design we have now an encoder that integrates all the patches spatially (for each timepoint). The confusing part for me is the predictor: on the one hand it tries to capture global features between all tokens in the time domain, on the other hand the alignment loss forces the tokens to only capture the local features at specific timepoint. What is the main motivation behind this? Note that from the experiments it is shown that the method works quite well, but it was confusing for me in the beginning to understand how did you come up with this design choice. Stated differently: how does the model perform if we do the alignment loss before the predictor, or if we remove the predictor.

How did you evaluate the TUAB dataset since your model can only work 4s windows. The TUAB dataset can have many minutes to hours of EEG data. Same goes for TUEV.

The evaluation for TUAB seems to not include in the comparison the best performing models in the literature (see limitations). Are there also other models for the other tasks that perform better and are not included in the paper?

**Limitations:**

The model is only trained with 4s of EEG data and the pretraining set is very small. Although the initial experiments are promising, It is not clear how the model generalises in other datasets and downstream tasks. The model is not evaluated how it performs if we finetune the model (instead of linear probing).

---

> ### Author Rebuttal · Authors · 2024-08-05
>
> We thank the reviewer for the constructive feedback. We have uploaded a revision and used blue to mark the new changes. Our detailed responses are as follows.
>
> #### **W1**
> We added more test results for more datasets in the ablation experiment and scale law experiment, see Revision Section 3.4 & Appendix A. Channel embedding relationships are computed based on channel embeddings extracted from the pretrained model and are independent of the downstream task.
>
> #### **W2**
> In the scale law experiment, we added test results for the TUAB dataset and the BCIC-2B dataset, see Revision Appendix A.4. The results on the TUAB dataset and the BCIC-2B dataset show that they present a scale law for the performance metrics with respect to the model size.
>
> We added pretraining experiments using 100%, 50%, 25%, and 12.5% of the training data and tested them on the downstream tasks of BCIC-2A and BCIC-2B, see Revision Appendix A.5. The results show that their performance metrics exhibit a scale law with respect to the amount of data.
>
> #### **W3&Q1**
> The design motivation of the predictor: (1) on the one hand, it predicts the representations of the masked "50% time patches" (equivalent to the BERT-style task), which allows the encoder to extract features in each local patches, that contribute to the task of completing masked patches in the time dimension; (2) on the other hand, it avoids the collapse of representation due to the direct alignment of the outputs from the encoder and momentum encoder (i.e., the parameters of the two are difficult to change during pretraining), and in this way, it is similar to the predictor in BYOL [1]. Also due to the design of (1) we can avoid predictor learning to be a identity map.
>
> We conducted a pretraining experiment after removing the predictor, the variation of $L_R$ loss with the number of iteration steps during training is shown in Figure 6 of the Revision Appendix A.1. In Figure 6, the $L_R$ loss of the model without the predictor does not decrease, which indicates that directly aligning the outputs of the encoder and the momentum encoder does lead to the problem of representation collapse, resulting in the model can not learn meaningful representation.
>
> #### **W4&Q3**
> (1) The difference in accuracy on TUAB is due to the different sample lengths used by the different methods. For a fair comparison, we use the same evaluation approach on TUAB as in the LaBraM and BIOT papers, which both use 10-second samples for classification. ChronoNet [2] uses a window of different sizes of at least 60s for classification and achieves an accuracy of 86%. In the second paper [3] it used 1-minute samples for classification. This shows that different sample lengths can have a huge impact on the experimental results.
>
> (2) Our aim is to develop better pretraining methods so that the models can achieve better performance (more generality) on a wide range of EEG tasks. Therefore on other tasks we only compare with similar pretrained models. On some tasks, there exist complex models that may achieve better results using handcrafted features or dedicated designs, but they are developed for specific tasks and may be more difficult to generalize across tasks compared to pretrained models.
>
> #### **W5**
> TUEG belongs to the clinical dataset and the data are related to patients and brain diseases, the channels of TUEG dataset are mostly 23 channels 10-20 standard electrodes, not diverse enough. We trained our model on a limited number of pretraining datasets, which includes more brain-computer interface related tasks such as motor imagery, error-related potentials, sleep stage detection, and identity recognition. These datasets also contain more channels. The TUEG dataset was chosen as the downstream task dataset in order to better compare with other pretrain models such as BIOT.
>
> Despite the abundance of labeled data for epilepsy and sleep task, many works [4] [5] [6] have used self-supervised methods to achieve performance that exceeds that of supervised methods.
>
> |Methods|Model Size|Balanced Accuracy|AUROC|
> |-|-|-|-|
> |Ours (no pretrained)|25M|0.7553±0.0014|0.8260±0.0018|
> |Ours|25M|0.7983±0.0030|0.8718±0.0050|
>
> We added test results using random initialization parameters on TUAB (see Appendix A.7), and according to our experimental results, even with a small amount of pretraining data, the pretrained model works better on the TUAB dataset compared to the unpretrained model.
>
> #### **W6**
> We added more detailed descriptions for the tables, see Revision PDF Section 3.4. In Table 5, we changed the headers to BCIC-2A-BAC, BCIC-2B-AUROC, and KaggleERN-AUROC. In Table 6, we update the header to BCIC-2A-BAC.
>
> #### **W7**
> We have fixed all spelling errors and regularized the citation reference format, see Revision PDF.
>
> #### **Q2**
> In Revision Appendix C.2.6 Table 12&13, we show the architecture of our model used in the TUAB and TUEV.
>
> **[ Model architecture for TUAB & TUEV ]**
>
> |In Size|Opt|kernel|stride|groups|padding|
> |-|-|-|-|-|-|
> |23xT|conv1d,norm,gelu|1|1|1|0|
> |20xT|conv1d,norm,gelu,dropout|K|1|20|K/2|
> |20xT|encoder|64|64|||
> |(T/64)x4x512|flatten,linear|||||
>
> The 23-channel input is first to reduce the number of channels to 20 by the convolution (K=15 for TUAB and 55 for TUEV). Then, the eegpt-encoder uses the 20 channel embeddings and maps 64-length window segments of the input signals to 4 (number of summary tokens)x512-dimensional features. Finally, the flatten and linear layers are used to output the final classification score.
>
> #### **L1**
> See author rebuttal ablation experiment.
>
> [1] Bootstrap your own latent-a new approach to self-supervised learning.
>
> [2] ChronoNet: A deep recurrent neural network for abnormal EEG identification.
>
> [3] Window Stacking Meta-Models for Clinical EEG Classification.
>
> [4] Channel-Aware Self-Supervised Learning for EEG-based BCI.
>
> [5] MAEEG: masked auto-encoder for EEG representation learning (2022).
>
> [6] BIOT: Cross-data biosignal learning in the wild.

---

### Author Rebuttal · Authors · 2024-08-05

We thank all the reviewers for your time and constructive feedback. During the rebuttal, we have prepared a revision and used blue to mark the new changes. Below are the results of the added experiments as the response.

**[Table 1: Ablation study for pretraining methods (Appendix A.2)]**

In the ablation experiment, we added test results for the BCIC-2B and KaggleERN datasets, as shown in the table below. The 'BAC' is the Balancing Accuracy, and more detailed descriptions of the metrics can be found in Revision Appendix D.

| Variants                 | $\mathcal{L}_{A}$ | $\mathcal{L}_{R}$ |       BCIC-2A-BAC |     BCIC-2B-AUROC |   KaggleERN-AUROC |
| ------------------------ | ----------------: | ----------------: | ----------------: | ----------------: | ----------------: |
| A: w/o $\mathcal{L}_{A}$ |             37.13 |              0.57 |     0.5287±0.0086 |     0.7264±0.0381 |     0.5752±0.0164 |
| B: w/o $\mathrm{LN}$     |              0.15 |             0.002 |     0.5567±0.0088 |     0.7920±0.0012 |     0.5891±0.0227 |
| C: w/o skip              |              0.12 |              0.56 |     0.5796±0.0011 |     0.7702±0.0122 |     0.6356±0.0296 |
| D: with all              |              0.24 |              0.56 | **0.5846±0.0070** | **0.8059±0.0032** | **0.6621±0.0096** |

The results of the ablation experiments with the addition of the dataset are still in line with the conclusions in the original paper:(1) the model performance decreases significantly on all datasets without the $L_A$ loss (6% to 9%); (2) the performance of the variant B model without layer normalization of the target patches decreases by 3%, 1%, and 7% on BCIC-2A, BCIC-2B and KaggleERN, respectively; (3) the performance of the variant C model with the removal of skip connection was reduced by 1%, 3%, and 3% on the three datasets, respectively.

**[Table 2: Ablation experiments of fine-tuning methods (Appendix A.3)]**

In the ablation experiments, we added experiments comparing the linear probing method with the full fine-tuning method, see Revision Appendix A.3, as shown in the table below.

| Variants | ASF  | L-P  | BCIC-2A-BAC       | BCIC-2B-AUROC     | KaggleERN-AUROC   |
| -------- | ---- | ---- | ----------------- | ----------------- | ----------------- |
| A        |      |      | 0.5774±0.0072     | 0.7871±0.0054     | 0.6078±0.0101     |
| B        | ☑️    |      | 0.5183±0.0155     | 0.7541±0.0083     | 0.6110±0.0019     |
| C        |      | ☑️    | 0.5586±0.0089     | 0.7974±0.0030     | 0.6463±0.0081     |
| D        | ☑️    | ☑️    | **0.5846±0.0070** | **0.8059±0.0032** | **0.6621±0.0096** |

In the above table, ASF denotes with adaptive spatial filter, instead of directly feeding the signal into the model; L-P denotes using linear probing, instead of using the full fine-tuning of the model. Model variants A and C are models with full fine-tuning and linear probing, respectively, after excluding the adaptive spatial filter. Model variants B and D are models with full fine-tuning and linear probing after using adaptive spatial filter, respectively. The results show that on the BCIC-2B and KaggleERN datasets, variants C and D tested with linear probing achieve better results than variants A and B using full fine-tuning; on the BCIC-2A dataset, variant A with full fine-tuning and no adaptive spatial filter is close to variant D with linear probing and adaptive spatial filter. Overall, the variants C and D (using linear probing) outperform A and B. Also, the results show that variants B and D with adaptive spatial filter tested on the BCIC-2A, BCIC-2B and KaggleERN datasets achieve better results compared to variants A and C without adaptive spatial filter.

---

### Decision · Program_Chairs · 2024-09-25

**Decision:**

Accept (poster)

**Comment:**

This paper develops a 10M pre-trained transformer model	for EEG
processing.  The transformer is trained with masked self-supervised
training with spatio-temporal representation alignment.  Results on
several down-stream tasks are improved relative to comparison
approaches.  All reviewers scored this with a 5 or 6 recognizing that
it is worth presenting to the NeurIPS conference.